

**A case study of field-scale maize irrigation patterns in Western Nebraska: Implications to**
**water managers and recommendations for hyper-resolution land surface modelling**
Justin Gibson[1], Trenton E. Franz[1], Tiejun Wang[1,2], John Gates[3], Patricio Grassini[4], Haishun
Yang[4], Dean Eisenhauer[5]
[1]School of Natural Resources, University of Nebraska-Lincoln
[2]Institute of Surface-Earth System Science, Tianjin University, Tianjin 300072, P.R. China
[3]The Climate Corporation, San Francisco, CA
[4]Department of Agronomy and Horticulture, University of Nebraska-Lincoln
[5]Biological Systems Engineering, University of Nebraska-Lincoln
Corresponding author J. Gibson (jgibson8@huskers.unl.edu)



**Abstract**

15        In many agricultural regions the human use of water from irrigation is often ignored or

poorly represented in land surface models and operational forecasts. Because irrigation increases
soil moisture, the feedbacks to surface energy balance, rainfall recycling, and atmospheric
dynamics are not represented and may lead to reduced model skill. In this work, we describe four
plausible and relatively simple irrigation routines that can be coupled to the next generation of
hyper-resolution LSMs operating at scales of 1 km or less. The irrigation output from the four
routines (crop model, precipitation delayed, evapotranspiration replacement, and vadose zone
model irrigation based) are compared against a historical field scale irrigation database (2008-
2014) from a 35 km$^2$ study area under maize production and center pivot irrigation in western
Nebraska (USA). Here we find the most conservative irrigation routine (crop model) produces
seasonal totals of irrigation that compare well against the observed irrigation amounts across a
range of wet and dry years but with a low bias of 80 mm yr$^{-1}$. The most aggressive water savings
irrigation routine (vadose zone model) indicates a potential irrigation savings of 120 mm yr$^{-1}$ and
yield losses of less than 3% against the crop model benchmark and historical averages. The
results from the various irrigation routines offer insights to local water managers about the
potential value of water savings technologies and irrigation practices. Moreover, the routines
offer the hyper-resolution LSM community a range of irrigation routines to better constrain
irrigation decision making at critical temporal (daily) and spatial scales (<1 km).
Keywords: Crop model; Irrigation; Water savings technology; Maize; Hydrus



## 1. Introduction

Regional land surface models (LSM) often ignore or do a poor job of representing irrigation physics (Kumar et al., 2015). This is in part due to the difficulty of validating irrigation amount estimates as irrigation datasets are rare, in formats that are difficult to work with on a regional scale (e.g., different reporting formats from one agency to another or in paper records), and have a latency period of months to years making them impractical to use in operational forecasts. The USDA produced Farm and Ranch Irrigation Survey (USDA, 2014) contains survey data on the county level, however data are only reported every five years and irrigation data are given on a pumping volume basis instead of depth per irrigated area as needed by LSMs (Siebert et al., 2010). Another well-known irrigation database, AQUASTAT (FAO, 2008), contains irrigation data at a spatial scale too coarse for investigating important feedbacks like land-atmospheric coupling and lacks information for Europe and North America. There are only a few studies that have used field-level irrigation databases (c.f. Grassini et al. 2011, 2014, 2015), mostly focusing on benchmarking on-farm irrigation in relation to crop production.

With the continual refinement in the spatial resolution of LSMs down to <1 km (Wood et al., 2011) and the coupling to crop models (Kucharik, 2003), reliable irrigation data need to be incorporated in the calibration and validation of LSMs. One area of particular importance is the impact of soil moisture on atmospheric processes, such as rainfall recycling (Findell and Eltahir, 1997), the strength of atmospheric coupling (Koster et al., 2004), and planetary boundary layer dynamics (Santanello et al., 2011), all of which impact the skill in operational forecast models. More complicating is that irrigation timing and volumes are based both on human decision making processes and biophysical requirements (Gibson, 2016). For example, the USDA found 24% of producers relied on crop calendars, 16% on crop consultants, and 23% on in-situ probe


technology (USDA, 2014). Because irrigation decisions are dependent on both processes,
reliable historical irrigation data are critical to understand why and how decisions were made in
order to accurately represent the physics in hyper-resolution LSMs and operational forecast
models. In the absence of irrigation data, LSMs have typically relied on mass balance approaches
(Döll and Siebert, 2002; Wada et al., 2012) where irrigation amounts close the water balance.
While a reasonable first approach, this methodology may introduce additional uncertainty into
LSMs due to the complexity of representing the human decision making process on water use.
The uncertain irrigation schemes affect the time history of soil moisture and thus our ability to
properly assess the impacts of human water use on coupled land-atmospheric model physics.
The focus of this study was to investigate historical irrigation use at the critical field scale
(~0.8 by 0.8 km) in a study area of 3500 ha in western Nebraska, which resides on the edge of
the USA Corn Belt. While a relatively small area, the study site is an ideal location for assessing
the sustainability of groundwater pumping for irrigation of crops. The study area is a microcosm
of many areas across the globe, where humans rely on groundwater withdrawals for their
livelihoods (Mekonnen and Hoekstra, 2011). The study area is at a critical location as it is on the
boundary where irrigation supply volumes can no longer economically compensate for the deficit
between potential evapotranspiration ($ET_p$) and precipitation ($P$). Of particular concern to
impacts on both human and natural ecosystems are the resultant declines in the local water table
due to irrigation (Young et al., 2014). For example, the southern portion of the High Plains
Aquifer (HPA) has had significant groundwater depletion over the last 80 years, with up to 50%
losses of saturated thickness (Scanlon et al., 2012). In the Northern HPA, where this study area is
located, intense irrigation pumping has led to localized water table declines (specifically in Box
Butte County, and widespread throughout the neighboring Upper Republican Natural Resources



District) but has yet to be widespread across the region (Young et al., 2013). Given low recharge
(Szilagyi and Jozsa, 2013; Gibson, 2015; Wang et al. 2016) relative to irrigation pumping, rising
global food and water demands (FAO, 2009), and concomitant effects of climate change (Kumar,
2012), the sustainability of this study area and the overall HPA system in support of long-term
irrigation agriculture is uncertain. The study presented here is an important first step in assessing
water saving technologies to continue to make irrigation agriculture sustainable for its critical
need in meeting rising global food demands.

Here, we benchmark relatively long-term (2008-2014) and field-specific flow-meter

measured irrigation amounts within the study area against a range of irrigation strategies. The
data includes information on 55 fields (~65 ha) producing maize under center pivot irrigation.
Datasets at this critical LSM scale are rare due to privacy concerns and as a result are often
aggregated to county and seasonal totals (USDA, 2014; USDA-NASS, 2014) making assessment
of the irrigation depths over a given area difficult to ascertain. This study therefore fills a critical
data need in the development and testing of the next generation of hyper-resolution LSMs and
operational weather forecast models. The next generation of LSMs will be essential for better
assessing the impacts of irrigation on the surface energy balance as well as evaluating the long-
term sustainability of groundwater resources in agricultural areas.

The primary objective of this study is to benchmark historical irrigation amounts in the

study area using different plausible physically based irrigation triggering regimes. In the methods
sections we will summarize the four identified irrigation triggering regimes- 1. crop model (CM),
2. Precipitation delayed (PD), 3. Evapotranspiration replacement (ET), and 4. Vadose zone
model where irrigation is triggered by simulated pressure head (H). In the results section we will
assess the impacts of annual variations in precipitation on irrigation, and soil texture differences



in the study area. In the discussion, we will provide a general framework for including plausible
irrigation schemes in LSMs, as well as discuss any expected changes in irrigation behaviors as
producers adopt various technologies into practice. The framework and irrigation schemes
provide LSMs a practical guideline for estimating irrigation depths and timing as well as a
strategy for investigating technology adoption scenarios.

**2. Methods**
**2.1 Description of Study Area and Historical Data**
The study area is located in western Nebraska where the South Platte River enters the
state (Fig. 1). The site encompasses 55 fields with an average area of 65 ha under irrigated maize
production (3500 ha total area). Overhead sprinkler irrigation from center-pivots using water
from the underlying HPA is the most common form of irrigation in this area as well as
throughout Nebraska, and the USA, as it is a cost effective and more efficient option than flood
irrigation. The study area is semi-arid where annual potential alfalfa referenced
evapotranspiration ($ET_r$) is significantly higher than precipitation ($P$) (HPRCC, 2016). The 7-
year (2008-2014) average annual $P$ is 440 mm/yr and average annual $ET_r$ is 1910 (mm/yr), as
measured by the High Plains Regional Climate Center weather station (HPRCC, 2016) located
within 10 km of the study area near Brule, NE.
Data obtained from SSURGO (Soil Survey Staff, 2016) indicates that soil texture in the
area falls within 2 USDA textural classes: sandy loam and loam (Fig. 2). Historical land
management data for the area are available from the South Platte Natural Resource District
(SPNRD, 2015). The SPNRD dataset includes field-specific information from the period of
2008-2014 on crop type, irrigation pumping volumes, and irrigated area. Detailed descriptions



and quality control of NRD databases can be found in Grassini et al. (2014) and Farmaha et al.
(2016). The above datasets provide the needed meteorological forcing, model parameters, and
calibration datasets for running and evaluating the suite of irrigation modeling routines described
below.

**2.2 Irrigation Modeling Routines**
In the following sections we will describe four identified irrigation triggering routines,
including CM, PD, ET, and H. The four irrigation triggering routines represent the upper limit of
irrigation requirements in which no plant water stress occurs (CM), and the lower irrigation limit
needed to ensure minimal yield loss against a crop model benchmark (H). Moreover, the four
routines can be easily coupled or implemented into LSMs. We also note the difference between
the historical irrigation practices and lower bound of simulated irrigation provides a potential
water savings value in the study area. This water savings value will be important for evaluating
the economics of new irrigation technologies as well as providing critical information to policy
makers and local stakeholders on the sustainable management of the HPA.

**2.2.1 Crop Model Irrigation (CM)**
A crop model, Hybrid Maize (HM) (Yang et al., 2013) was utilized to estimate irrigation
requirements and yield potential under an idealized scenario of crop growth with no water stress.
Model performance has been extensively validated against measured yield in crops that received
near-optimal management across the Corn Belt (Grassini et al, 2009, 2011). However, it has not
been rigorously tested for seasonal irrigation totals, which is one key outcome of this study.
Details on the model can be found in Yang et al. (2013) and a brief description of the model is



given here. Inputs to this model include meteorological data, soil texture, crop biophysical
parameters, sowing date, and plant density. The datasets are described above in section 2.1. Soil
water dynamics over the root zone are simulated through a bucket model approach with 10 cm
deep layers.  Drainage between soil layers occurs when soil moisture exceeds field capacity.
Irrigation application is triggered when actual ET ($ET_a$) is less than crop referenced potential
evapotranspiration ($ET_c$), ensuring no water stress occurs throughout the entire growing season.
Irrigation depth is determined by the amount of water needed to bring the profile back up to 95%
of field capacity. Maximum water application per irrigation event was set to 19.5 mm. When the
depth-weighted unsaturated hydraulic conductivity ($K_r$) of the root zone is greater than or equal
to $ET_c$, $ET_a$ is equal to $ET_c$. Otherwise $ET_a$ is equal to depth-weighted $K_r$ of the root zone.

**2.2.2 Precipitation Delayed Irrigation (PD)**

Water application in an idealized land management operation would consider all

components of the water balance within the decision making process. However, in practice,
precipitation is often the only component considered due to 1) the difficulty of accurately
measuring the other water balance components and 2) the relative economic return is minimal
when considering the perceived potential of crop yield loss versus savings due to reduced
pumping/irrigation. With this in mind, producers often develop "rules of thumb" to irrigate up to
a target total amount water equal to irrigation plus in-season rainfall (in the study area, 1 May to
30 September). Using these basic rules of thumb and local crop calendar requirements, we
suggest the following routine based off of precipitation data alone. However, we note that this is
not a recommendation for producer adoption, but instead represents a simplified method of
irrigation management for modeling purposes. In addition, the applicability of this method to



other regions should be possible with complimentarily datasets (i.e. $P$ and $ET_c$).
Recommendations obtained from the SPNRD indicate that maize requires approximately 650
mm of total water (precipitation plus irrigation, $P+I$) per growing season
(http://www.spnrd.org/index.html). Field observations indicate that irrigation often starts around
mid-June and concludes around mid-September, leading to a 100-day irrigation season. Average
irrigation application in the absence of precipitation would be 6.5 mm/day or 19.5 mm per 3 day
period. This irrigation depth is consistent with producer interviews and local expert knowledge.
Three day periods are critical to consider as this is often the time required to perform a single
$360^{\circ}$ rotation of a center-pivot. In this routine, if rainfall is greater than 6.5 mm/day, then
irrigation for one day is met, and thus a 1 day delay is set. Likewise, for a rainfall event of 13
mm/day, then two days of irrigation are met and irrigation is delayed 2 days, and so on for larger
rain events. For simplicity, rain events and irrigation delays are rounded to the nearest day and
up to a maximum of 7 days' delay. For rainfall events greater than 45.5 mm/day, we assume a
maximum delay of 7 days due to deep drainage and runoff losses incurring during the event.

**2.2.3 ET Replacement Irrigation (ET)**

The primary purpose of irrigation is to ensure $ET_a$ is able to adequately keep up with $ET_c$

over the growing season as $ET_a$ is linearly correlated with yield (Passioura, 1977). Proper
management allows a deficit between applied water and $ET_a$ in order to allow for adequate
infiltration after rainfall. This deficit was assumed to be 6.5 mm for this routine based on the
average daily crop water requirement discussed above. In this algorithm whenever the deficit
was greater than 6.5 mm during the irrigation season (15 June to 30 September) an irrigation



event of 19.5 mm was trigged for the next day.  Again, an irrigation event of 19.5 mm was used
as it represents a 3 day period, over which the center-pivot operates.

Estimating $ET_c$ is necessary in order to track the deficit between applied water and $ET_a$.

While estimating $ET_c$ is complex given the variability of micrometeorological variables from one
field to another, in practical applications, crop coefficients are often used to surmise the
differences in crop biophysical relationships and the effect of soil (Shuttleworth, 1993). These
coefficients are often published from local services like the state climate office or HPRCC in
Nebraska.

Here, $ET_c$ (mm/day) was estimated following the single crop coefficient method outlined

in Allen et al. (1998):
$$ET_c = ET_r\, K_c \tag{1}$$
where $ET_r$ (mm/day) is reference crop $ET_p$ calculated from micro-meteorological variables, and
$K_c$ is a dimensionless empirical constant that encompasses crop development as well as the
average effect of soil on evaporation rates. Daily $ET_r$ data were determined from the HPRCC
weather station data. $K_c$ values were calculated as a function of growing degree day
accumulation ($GDD$) from the HPRCC data (HPRCC, 2016). A single day calculation of
growing degrees ($GDD_{daily}$) is defined as:
$$GDD_{\text{daily}} = \frac{T_{\max} + T_{\min}}{2} - T_{base} \tag{2}$$
where $T_{\max}$ is the daily maximum temperature ($^o$C) (with a maximum of 30$^o$C), $T_{\min}$ is the daily
minimum temperature ($^o$C), and $T_{\text{base}}$ is 10$^o$C. The $GDD$ method is preferred as it more
accurately represents a proxy for crop development, as opposed to a fixed number of days after
sowing.





### 2.2.4 Hydrus-1D Irrigation (H)


A physically based vadose zone model, HYDRUS-1D (H1D) (Šimůnek et al., 2013) was
used to simulate irrigation requirements based on predefined soil pressure head trigger points in
the root zone. In order to carry out necessary seasonal dynamics for annual crops (i.e. dynamic
root growth, root distribution), we coupled the HM and H1D models using Matlab. We note that
soil pressure triggered irrigation events based on more than one soil pressure value, flexible
irrigation timeframes, and dynamic root growth with a specified distribution are unavailable in
the standard H1D code. Here we use Matlab to link together a series of one day simulations
(totaling 7 years), where model outputs (pressure head at depth, flux rates, actual
evapotranspiration, etc.) at the end of the day were used to make a decision about irrigation for
the following day.
H1D simulates soil water dynamics and water flow by a numerical approximation to the
1D Richards equation:
$$\frac{\partial \theta}{\partial t} = \left(\frac{\partial}{\partial z}\right)\left[K(\theta)\left(\frac{\partial h}{\partial z}+1\right)\right]-S \qquad (3)$$

where $\theta$ is volumetric water content ($cm^3/cm^3$), $t$ is time (day), $z$ is the spatial location
(cm), $K(h)$ is unsaturated hydraulic conductivity (cm/day), $h$ is pressure head (cm), and $S$
is a sink term describing evapotranspiration (1/day). The soil profile simulated is 6 m
deep with 1 cm node discretization.  Free drainage is set for the lower boundary
condition, as local depth to groundwater is on average 15 m (Korus et al., 2013)
The H1D model requires $ET_c$ be partitioned into potential evaporation and potential
transpiration. This is accomplished using Beer's law:
$$T_p = ET_c \left(1-e^{-k*LAI}\right) \qquad (4)$$



$\quad E_p = ET_c - T_p$ $\qquad\qquad\qquad\qquad\qquad\qquad\qquad\qquad$ (5)
$\quad$ where $T_p$ is potential transpiration (cm/day), $E_p$ is potential evaporation (cm/day), $k$ is the light
$\quad$ extinction coefficient (set here to 0.55 (Yang et al., 2013)), and $LAI$ (m$^2$/m$^2$) is the leaf area
$\quad$ index. We simulated one multi-year $LAI$ seasonal dynamic using HM. This same seasonal
$\quad$ dynamic was used for all simulations. In addition, HM was used to estimate date of silking for
$\quad$ each simulated year. Water stress is minimized during silking periods as this is the most critical
$\quad$ grain filling period for yield. Most producers will heavily water in this period to ensure yield. In
$\quad$ order to accurately represent the irrigation behavior, we forced irrigation events every three days,
$\quad$ one week before and after the silking date. In the case where a simulated day occurred during the
$\quad$ growing season, root depth ($Zr,$ cm) and root distribution ($Zr_{RD}$, dimensionless) parameters were
$\quad$ calculated on a daily basis based off of a pre-determined $GDD$ accumulation after planting date
$\quad$ for each growing season. This process was carried out following the equations outlined in the
$\quad$ HM user manual (Yang et al., 2013):
$\quad Zr = \dfrac{GDD}{GDD_{\text{Silking}}} Zr_{\max}$ $\qquad\qquad\qquad\qquad\qquad\qquad$ (6)
$\quad Zr_{RD} = \exp(-VDC\, Z_L / Zr)$ $\qquad\qquad\qquad\qquad\qquad\qquad$ (7)
$\quad$ where $GDD_{silking}$ is growing degree days at silking, $ZR_{max}$ is a biophysical parameter representing
$\quad$ the maximum depth the root zone can reach (cm) and set to 150 cm here (Yang et al., 2013),
$\quad$ $VDC$ is a vertical distribution coefficient set to 3 here, and $Z_L$ is the current depth in the root zone
$\quad$ (cm). In addition, HM was used to estimate date of silking for each simulated year.
$\qquad$ Irrigation events and depths for the following day were calculated by investigating the
$\quad$ average soil pressure heads at 30, 60, and 90 cm during the historical irrigation period from June
$\quad$ 15 through September 30. Prior to the silking date, the average soil pressure head at 30 and 60



cm is computed and compared against a preset irrigation trigger value set to -500 cm based off of
the dominant soil types in the area (Fig. 2). Following the silking date, the average soil pressure
is computed at 30, 60, and 90 cm with the same trigger point of -500 cm of pressure. This
algorithm is based on best practice irrigation recommendations summarized in Irmak et al.
(2014). In practice, producers vary the irrigation pressure trigger point based upon farmer risk
aversion and soil type. Given that yield is the primary economic driver over energy costs for
pumping water, this trigger point is often set at conservative values. When the pressure head at
the considered depths exceeds the trigger point, an irrigation event of 19.5 mm is set for the
following day. The irrigation event is added to any precipitation that may arrive randomly on that
day as well.

In order to numerically advance the models through time, we set up a series of 1 day

simulations and logical statements. If the model date occurred outside of the growing season
(October 1 to April 30), no changes were made to precipitation and bare surface was simulated.
If the model day was after planting (1 May) and before the start of the historical irrigation season
(15 June), only the root zone depth and root distribution parameters were updated. For model
dates during the irrigation season (15 June to 30 September), the root zone depth, root
distribution, and irrigation amounts were changed for the following day. Using this routine, the
model was run continuously at 1 day intervals for the entire study period (1 January 2008 to 31
December 2014).

**2.3 Rainfall Variability Across the Study Site**

Daily precipitation data for the years 2008-2014 were available from 7 gauges within a

radius of 35 km of the study site. In order to help assess the effect of precipitation variability on



irrigation application, all 7 time series along with the average precipitation time series were used
within the four irrigation routines described above. In addition, all irrigation routines that
considered soil type were repeated for the two dominant soil types in the study area, i.e., sandy-
loam and loam.

**3. Results**
**3.1 Precipitation Variability and $ET_c$**

As expected, significant gauge-to-gauge variability was observed within the 7 rain gauge

time series within each growing season with a mean of 320 mm and a CV of 35% (Fig. 3).  In
general, as precipitation totals increased, the range in seasonal totals increased as well (slope =
0.246 mm yr$^{-1}$, $R^2$ = 0.38). There was no consistent year-to-year spatial precipitation gradient,
and no gauge consistently reported high or low totals. We hypothesize that this natural variability
in rainfall is a large contributor of the irrigation variability we see at the field level. This
hypothesis was beyond the scope of the current paper but suggest future research in this area.  In
terms of growing season $ET_c$, the HPRCC reported an average of 815 mm, and was within 10%
of county-level values estimated by Sharma and Irmak (2012).

**3.2 Historical Field Scale Irrigation**

Average seasonal irrigation over the 2008-2014 period was 380 mm with a CV of 23%.

Distributions of irrigation amounts are provided in the box and whisker plots given in Fig. 4.
Normal distributions and non-normal distributions with both negative and positive skewing were
observed (D'Agostino-Pearson test, $p<0.05$). Growing season precipitation plus irrigation
averaged 700 mm (Fig. 5) with a CV of 5%. The highest seasonal irrigation average occurred





during the growing season of 2012 (580 mm) due to an extremely dry growing season with only
80 mm of rainfall.  We found that soil texture was not a significant factor affecting irrigation
application at the field scale in this region. After grouping the fields by soil type (loam and
sandy-loam), we found that the mean irrigation for all years were not statistically different from
each other (Student's t-test, $p = 0.73$). This indicates that soil type did not factor into the
irrigation decision making process.

**3.3 Comparison of Historical Seasonal Irrigation Amounts with Four Irrigation Routines**

Results of the comparison between the historical irrigation (2008-2014) and the four

irrigation routines are summarized in Fig. 6. Both the CM and PD routines reproduce irrigation
amounts near the historical average.  CM irrigation water requirements were on average, 80 mm
lower (20% of total) relative to historical irrigation. For PD, the average seasonal difference was
40 mm lower (10% of total). For ET and H, simulated irrigation amounts were 80 mm (18% of
total) and 120 mm (30% of total) lower than the historical average, respectively. We also note
the slopes of the observed irrigations and the CM and PD for the given years were in general
similar. However, it is obvious from Fig. 6 that the slopes of ET and H were different from the
observations, which results in larger deviations in drier years and thus a potential for greater
water savings. The implications to water management will discussed in the next section.

**3.4 Irrigation Sensitivity to Rainfall**

All irrigation regimes responded to differences in the eight rainfall time series, and this

response is represented as vertical error bars in Fig 5. The difference between the highest and
lowest irrigation amount for each growing season was on average 75 mm, or 20% of average





irrigation totals. The largest difference in irrigation totals occurred in 2008 for all irrigation
regimes with an average of 130 mm between all 4 routines, and the smallest difference occurred
in 2012 at an average of 27 mm due to uniformly low precipitation. The analysis illustrates the
variation in irrigation amounts depends on which rainfall gauge is used to make a decision.
Given that producers often have fields distributed across a region the uncertainty in local rainfall
directly propagates into variations in irrigation amounts. Future research efforts should
investigate the effect of spatial rainfall variability on producer decision making but this was
beyond the scope of the current study.

**3.5 Soil Texture impact on Irrigation Routines**
We found that the two dominant soil textures in the study area did not have a significant
impact on irrigation amounts under CM and H. In the case of CM, average irrigation was within
1% for all years. For H, the irrigation average of the sandy loam soil was 10% less than the
average of the loam soil. Soil hydraulic parameters used for both soil textures were determined
using ROSETTA (Schaap et al., 2001) and are presented in table 1.

**3.6 Simulated Yield under Irrigation Routines**
Following the simulated irrigation for the routines of PD, ET, and H, the ($P+I$) time
series were reinserted back into the crop model for all years to estimate yield impacts (Fig. 7).
The crop model yielded an average 14.6 Mg/ha over the study period. The yield gap (i.e.,
difference between yield potential and actual yield) of US irrigated maize represents
approximately 15% of the potential (Grassini et al., 2013, http://www.yieldgap.org/), suggesting
an average actual yield of 12.4 Mg/ha for the study area, which is within 5% of historical





reported yield. For the three routines and for all years, simulated yields were on average within
97% of the simulated yield based on the CM. The results indicate that the various irrigation
scheduling strategies did not have a large impact on yield while reducing irrigation amounts
substantially; hence, they may be a sound economic decision for producers.

**4. Discussion**
**4.1 Temporal Variability of Applied Irrigation**
Historically, the study area has had a consistent amount of total seasonal water (P+I)
from year to year. The percent of irrigation to applied water (I/(P+I)) was on average 55%, and
notably in 2012 this was as high as 88%. The relative weight of irrigation to precipitation
highlights the importance for constraining irrigation amounts for proper water balance closure
within the study area, as well as in other areas with intense irrigation application. Given the high
seasonal rates of irrigation to precipitation, no doubt the soil moisture will be adversely affected
when compared to a rainfed area. More importantly, the impacts to the local surface energy
balance (Santanello et. al, 2011), rainfall recycling, and skill in observational forecasts may be
diminished without proper accounting for irrigation. For example, regional mesoscale modelling
illustrated that up to 40% of East African annual rainfall can be attributed to irrigation across
India (Vrese et al., 2016). With the suggested findings here on reduced irrigation needs (up to
30%), the potential changes to precipitation patterns across the HPA due to adoption of irrigation
scheduling technology should be further investigated.
The study area is currently under ground water appropriation, with a historical increase in
depth to groundwater of 1.2 m over the period of 1971 to 2013 (SPNRD, 2013; Young, 2013).
Precipitation pattern changes in the area induced by global warming are believed to lead to less



frequent but more intense storms with an increase in total precipitation (Dai et al., 2011).
However, the timing of precipitation is of equal concern to totals, as more infrequent rain events
may still lead to increased pumping with the same seasonal totals. The scenario of changing
precipitation amounts and timing is not unique to the study area but a more general pattern of the
region, highlighting the need for explicit treatment of irrigation depths and timing to fully
understand the complex feedbacks that exist beneath the land surface and atmosphere. The
irrigation routines suggested in this work can be used as a first assessment of the likely irrigation
amounts due to different observed scheduling practices (USDA 2014).

**4.2 Spatial Variability of Applied Irrigation**

The rainfall sensitivity analysis demonstrated the affects and uncertainty for each of the

four irrigation routines investigated. Lower rainfall years had lower spatial variability and as a
result simulated irrigation for each routine led to similar values. However, this behavior was not
consistent with the observed irrigation data, in which the lowest rainfall year (2012) had the
largest standard deviation (168 mm) for applied irrigation. The results are likely due to two
reasons: 1) producers give up irrigation at some point during the growing season as their crop
parishes in the extreme heat and drought conditions and 2) differences in well-to-well pumping
capacity become more apparent with increased pumping demand. Although no direct work has
been done to confirm differences in pumping capacity or inefficiencies in the study area, the
general effect has been explored through modeling in other areas (Foster et al., 2014). With
respect to LSMs, these two factors represent significant deviations away from water balance
closure approaches, making it challenging to include realistic irrigation values in dry years.





Therefore, additional studies and datasets similar to what is presented here are critical for the
calibration and validation of the next generation of hyper-resolution LSMs.
With regard to soil texture differences in the study area, observed irrigation data indicated
no difference between fields in these two texture classes. Similar behavior was seen from the
irrigation routine simulations that showed 10% difference for H and 1% for CM. We note given
the soil texture classes (and thus soil hydraulic parameters) this result is not unexpected. In
reality, we note the spatially varying techniques like variable speed and variable rate irrigation
are becoming increasingly popular and cost effective (Hedley and Yule, 2009; Hedley et al.,
2013). The small features within a field (e.g. sandy or gravelly areas, underperforming parts of
the field, water ways, pivot roads, etc.) can be better managed with the technology. Therefore,
managing fields following 1 dominant soil type (i.e. irrigation-pressure trigger point) may be
highly inefficient (Kranz et al., 2014). More refined and consistent soil texture data across
arbitrary political boundaries (Chaney et al., 2016) are needed to better account for differences in
irrigation water application on the sub-field scale, especially in areas with increasing adoption of
precision agriculture technology.

**4.3 Potential for Reduced Pumping**
The four irrigation routines presented represent different levels of allowable water stress
to develop in the maize. The CM routine is the lowest risk approach with respect to yield and
represents the modeled upper limit of required irrigation to maintain a stress free management
scenario. It is hypothesized that any irrigation application above this represents irrigation
application due to risk aversion, and will not appreciably increase yield. Comparisons between
2008-2014 indicate that the slope of the applied irrigation from observed irrigation are




indistinguishable, but with a bias of ~80 mm yr$^{-1}$ more observed irrigation. This indicates that
producers are averaging an additional 3-4 irrigation cycles beyond what the CM indicates. The
differences in irrigation totals from the other three irrigation routines are the result of increasing
allowable water deficit in the routines. A reduction of 115 mm or 30% of irrigation was observed
for the H when compared to the historical average.  We note this hypothetical scenario requires
perfect management, with full trust of the technology, and may not be achievable in practical
applications. However, we anticipate that a 50-75 mm reduction over a short technology
adoption period (2-4 years) is feasible, particularly in areas with strong university extension
programs and/or producer to producer knowledge exchange (Irmak et al. 2012). In addition,
these hypothetical reduced pumping numbers may be useful to local, state, and federal policy
makers about future water management decisions and investment in cost-sharing technology
programs.

**4.4 Assessment of Center-Pivot Irrigation Routines in Hyper-Resolution Land Surface**
**Models**

The four irrigation routines although biased, capture year-to-year variation in irrigation in

Western Nebraska. We believe the routines combined with a reasonable bias correction could be
easily incorporated into future hyper-resolution LSMs with the above routine descriptions and
readily available LSM model output or datasets. Additionally, the four routines could be run
offline in order to provide reasonable guesses of applied irrigation for a given irrigation season.
Finally, the four routines provide reasonable irrigation bounds and more importantly decreases in
irrigation as technology is introduced and adopted in particular areas.



**5. Conclusions**


In this work we describe four plausible and relatively simple irrigation routines that could
be coupled to the next generation of hyper-resolution LSMs operating at scales of 1 km or less.
The crop model irrigation outputs reproduce the year-to-year variability of the observed
irrigation amounts with a low bias of 80 mm yr$^{-1}$. Predications from the vadose zone model
indicate potential irrigation savings of up to 120 mm yr$^{-1}$ for maize. In addition, daily
precipitation variability across the study area was found to introduce significant variability in
daily irrigation decision making depending on which value was considered. Findings from the
work are useful to local water managers and stakeholders in evaluating potential water saving
technologies. In addition, the simple routines could be coupled to future hyper-resolution land
surface models that seek to understand the degree of land surface atmospheric coupling and
consequences to operational forecasts. This understanding is essential as society continually
recognizes the importance of human activities on the global water cycle and invests more
resources to understand the water-food-energy nexus.

**Acknowledgments**
This research is supported financially by the Daugherty Water for Food Global Institute at the
University of Nebraska. Access to field sites and datasets is provided by The Nature
Conservancy, Western Nebraska Irrigation Project, and South Platte Natural Resources District.
A special thanks to Jacob Fritton for critical insights into producer practices in the study area.
TEF would like to thank Eric Wood for his inspiring research and teaching career. No doubt the





skills TEF learned while at Princeton in formal course work, seminars, and discussions with Eric
will serve him well in his own career.

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





**Figures and Table**

Fig. 1: Study area located in western Nebraska with each field in the data set outlined.

Fig. 2: Area-weighted soil texture of all fields plotted on the USDA soil texture triangle, falling
primarly in the sandy loam and loam textures.

Fig. 3: Cumulative in-season precipitation measured at of 7 rain gauges and crop referenced
evapotranspiration ($ET_c$) calculated from a weatherstation <10km away. Precipitation variability
tends to increase with incresing seasonal totals.

Fig. 4: Box and whisker plots of historical irrigation depths. Upper and lower boundaries of
boxes indicated 75th and 25th percentile, respectively. Horizontal line within boxes is the median
value. Whiskers are maximum and minimum values. Asterisks indicate that irrigation
distribution deviates from a normal distribution (D'Agostino-Pearson test, $p<0.01$).

Fig. 5: Historical irrigation vs. the four simulated irrigation routines, for sandy loam (left) and
loam (right). Verticle error bars are standard error of the mean from the precipitation sensitivity
ananlysis and horizontal error bars are standard error of the mean from observed irrigation.

Fig. 6: Growing season totals for precipiptation (P), irrigation (I), and P+I. The dashed line
represents the historical average for P+I.

Fig. 7: Potenital yield simulated by Hybrid-Maize using the 4 irrigation routines: crop model
(CM), precipitation delayed (PD), evapotranspiration replacement (ET), and Hydrus-1D (H).


Table 1: Van Genuchten parameters used in Hydrus-1D simulations.



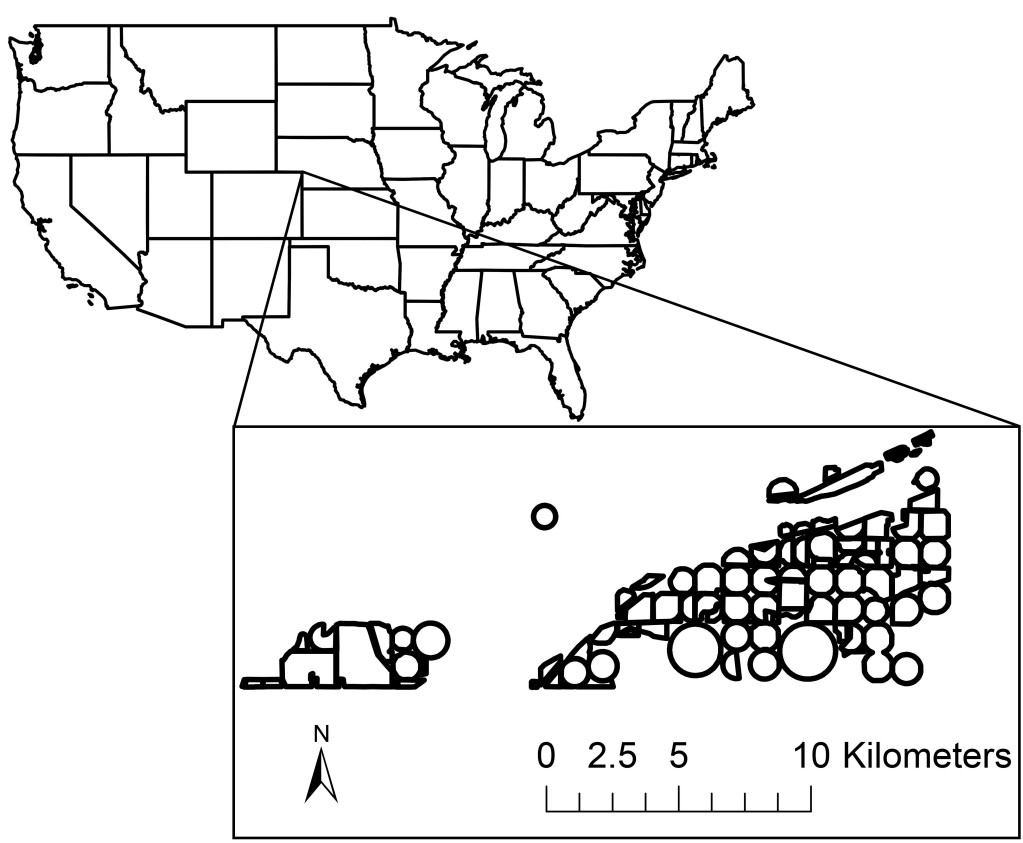

Fig. 1: Study area located in western Nebraska with each field in the data set outlined.





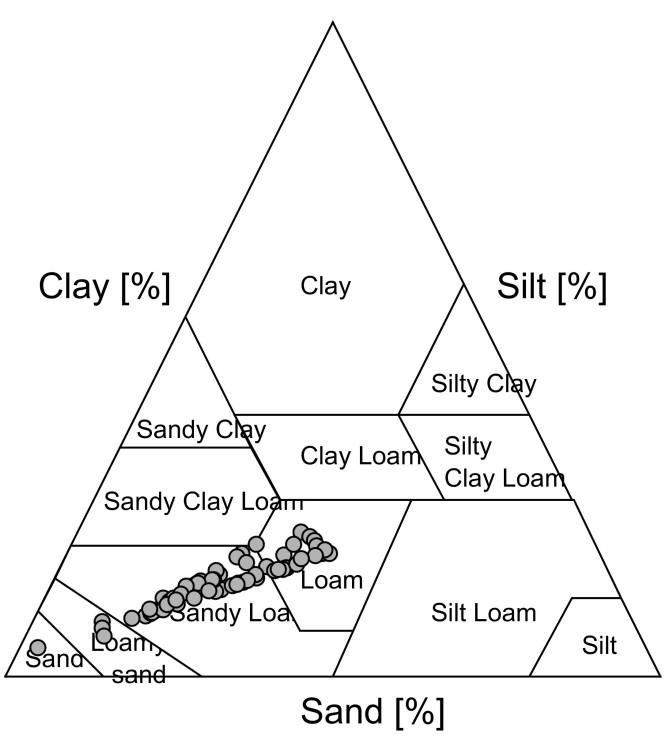

Fig. 2: Area-weighted soil texture of all fields plotted on the USDA soil texture triangle, falling
primarily in the sandy loam and loam textures.





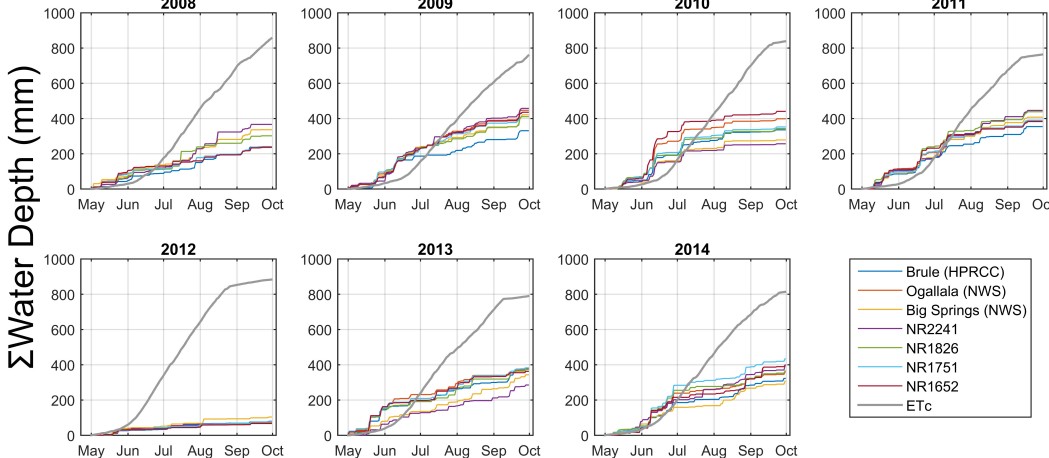

Fig. 3: Cumulative in-season precipitation measured at of 7 rain gauges and crop referenced
evapotranspiration ($ET_c$) calculated from a weatherstation <10km away. Precipitation variability
tends to increase with incresing seasonal totals.





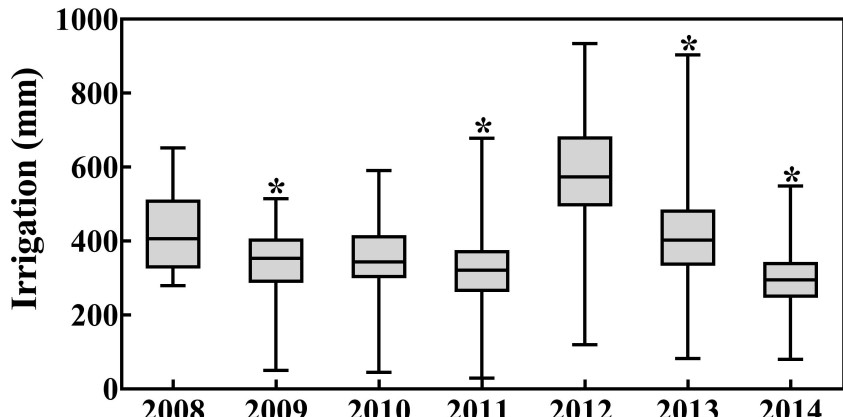

Fig. 4: Box and whisker plots of historical irrigation depths. Upper and lower boundaries of
boxes indicated 75th and 25th percentile, respectively. Horizontal line within boxes is the median
value. Whiskers are maximum and minimum values. Asterisks indicate that irrigation
distribution deviates from a normal distribution (D'Agostino-Pearson test, $p < 0.01$).





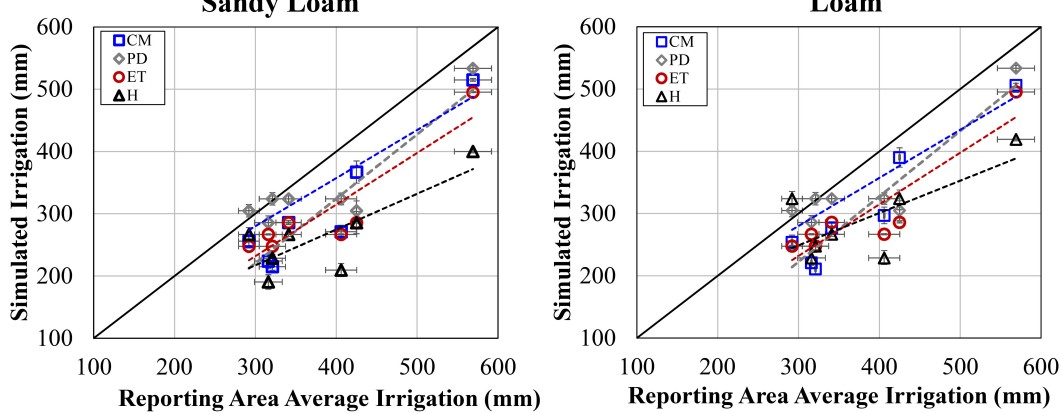

Fig. 5: Historical irrigation vs. the four simulated irrigation routines, for sandy loam (left) and loam (right). Verticle error bars are standard error of the mean from the precipitation sensitivity ananlysis and horizontal error bars are standard error of the mean from observed irrigation.





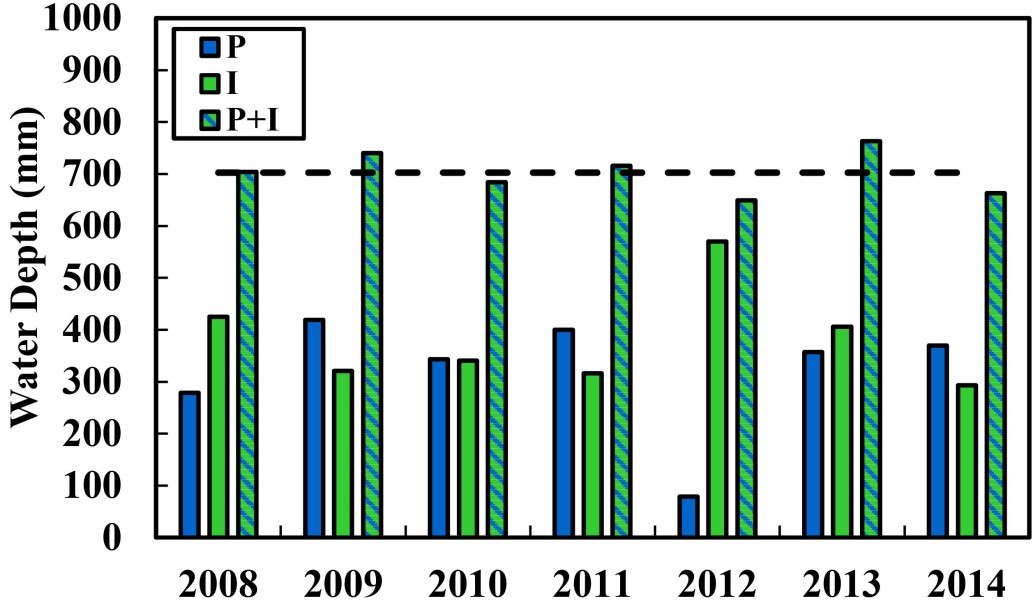

Fig. 6: Growing season totals for precipiptation (P), irrigation (I), and P+I. The dashed line
represents the historical average for P+I.





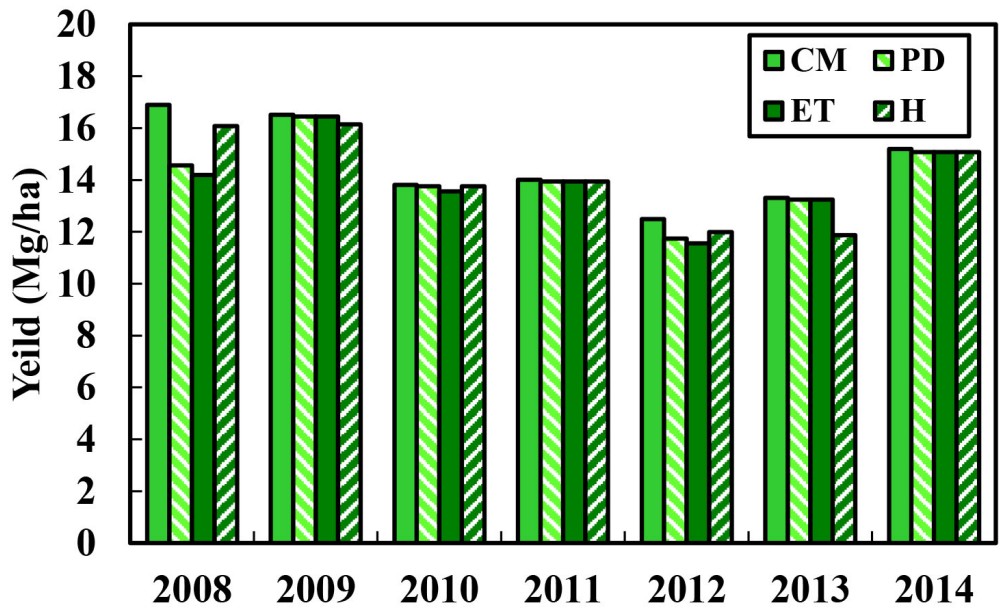

Fig. 7: Potenital yield simulated by Hybrid-Maize using the 4 irrigation routines: crop model
(CM), precipitation delayed (PD), evapotranspiration replacement (ET), and Hydrus-1D (H).





Table 1: Van Genuchten parameters used in Hydrus-1D simulations.

| Texture | $\theta_r$ (-) | $\theta_s$ (-) | $\alpha$ (1/cm) | $n$ (-) | $K_s$ (cm/day) |
|---|---|---|---|---|---|
| Sandy Loam | 0.048 | 0.385 | 0.0289 | 1.389 | 31.91 |
| Loam | 0.060 | 0.400 | 0.0127 | 1.458 | 10.85 |
