# Peer review of "A case study of field-scale maize irrigation patterns in Western Nebraska: Implications to"

_Hydrology and Earth System Sciences, 2016_

## Referee Comment (RC1) · Anonymous Referee #1 · 20 Sep 2016

Overall Comments:

Overall, this is a well-written manuscript describing 4 new ways to account for irrigation that could be used by managers and modelers alike. This type of work is much needed, as the human element/drivers of new LSM physics remain a challenge in how to account for them and prescribe them accurately. This is also a novel dataset put to good use. The schemes use sound assumptions and represent an array of complexities.

The paper is a worthwhile contribution, but becomes a bit thin in the results section and a few of the major limitations are glossed over and require further discussion. As

a result, I recommend major revisions in order to help the manuscript become more impactful and useful for irrigation-related studies.

In addition, I strongly recommend that, if possible, the results/analysis be extended to time series and sub-annual breakdowns of irrigation water vs. precipitation (and variability) for each of these schemes. Much of the utility for managers and more so for modelers will be on the diurnal and sub-seasonal scales, in which they need to obtain the water balance, soil moisture, and fluxes correct in order to couple to the atmosphere and represent the precipitation connection more accurately (i.e coupling).

Also missing is the broader applicability of these schemes outside of this unique, well-instrumented and reported-on field/domain. Other locations with less decision-making data points or coarser precipitation will no doubt find greater challenges.

Specific Comments:

L24: What is difference between a conservative and water savings routine? Sounds similar if you do not know the terminology. This is explained better in the paper itself, but maybe a word or two in the abstract could help better clarify what is meant by each.

L29: Is the actual transition of information and decision making part of this paper? Or is it suggested that it would be valuable in the future for managers? If the latter (which according to the paper itself there is no transition or decision making taking place (yet!), then please clarify this here to suggest it may be useful in the future (not that it already has been useful).

L52: might want to mention that the impact of SM on these is really modulated by the flux contribution to the atmosphere (SHF, LHF, or evap fraction, or just ET). So getting the SM-Flux relationship correct is critical, and i.e irrigation is essential as a component of that.

L58: Which are the 'both' here? L59: Is there a predictive nature to irrigation decision-making? Do Calendars vs. Consultants vs. Probe percentages change over time due

to other factors (technology, financial, drought, etc.)? Are consultant-based decisions consistent (is the advice consistent) over time?

L99: Not clear what is meant by 'irrigation triggering regimes'? Earlier (abstract) they were referred to as 'routines' that could be incorporated into LSMs. Regimes suggest something different?

L122: What is the native resolution of SSURGO relative to the study area and field scale?

L125: Same for SPNRD.

Section 2.2: Based on the descriptions of these, are this ranging from the most simple to most complex (in order)?

For H, would it be possible that the minimal yield loss could be set so high as to represent larger irrigation than in CM?

L168: 'amount of water'

L179: Has this approach been used in the past? There are no references, and based on interviews and expert knowledge. How did you come up with 6.5 exactly? If the ultimate goal is to have this in an LSM, I can envision that it might be very sensitive to this 6.5 number and thus overly simplistic. Are there any other knobs to turn?

L185: This sounds reasonable as first order approximations for extreme rainfall. What about the low-intermediate rainfall conditions and the speed of drainage? Should the delay estimates be constant regardless of the soil type (conductivity), land cover, and precipitation rate?

L243: What is meant by seasonal dynamic?

L280: All assumptions embedded in these approaches have been explained and seem reasonable. The proof is in the pudding, of course, and the results will bear that out. However, it might be useful to summarize what the input requirements and the assumed/tunable parameters are for each approach as well, if they are to be used in LSMs. An example here is the date ranges that are used. 6.5 is another as is -500cm, and the depths of the soil pressure.

L284: Where are they located with respect to the study site and the fields? Should some kind of interpolation (or average) be used as well?

L293: Mean ETc?

L294: Totals of what?

L297: This is critical. The 4 schemes rely on P as the most important input (right?). Forcing for LSMs comes from satellite and gauge-based datasets, likely much coarser (e.g. .125-deg) than the <1km field scale. How will this be addressed? How can we capture the irrigation variability without knowing that of Precip?

L325: I think a lot more could be said - this is the critical result/figure from this paper. There is a lot of error bar info on there and other aspects that could be discussed. The low bias stands out and is significant.

L328: 'Regimes' again.

L338: See earlier comment. This is a major limitation to all of these approaches and modeling irrigation at this scale.

L356: What does this imply about the assumed yield-irrigated amount relationship? That they underestimate and still didn't impact yield is even more surprising. There must be a lot of leeway (i.e. overwatering?).

L373: You are saying that, based on these models, you can get away with much less water and still produce the same yield, correct? Isn't that something that should have been quantified in the past (or known by the farmers)? Or is this still largely unknown? How certain are we that the models are correct and that the yield will still be met?

L384: Supports the need for a bit further analysis/figures looking into the time series

of the results.

L392: This was alluded to in an earlier comment: How can we know that prior decision making holds in the future or during other conditions not in the recent historical record?

L404: Why? Is it because soil types here are so are similar, with slowly varying properties?

L433: How about a controlled experiment/field to test sensitivity and realism of these schemes and resultant quantities? Is that reasonable in the future?

L443: Any predictive capabilities?

Section 4: This discussion section was welcome - lot of areas that need study but this is a good start.

L453: 'may be useful'?

Section 5: The conclusions are a bit thin, and perhaps should focus on some of the limiting factors and broader/future applicability (precip forcing, decision making, soil properties).

Fig. 1: Hard to tell exactly where these fields are as this box points to a point on the corner of CO and NE.

Fig. 1: Might be interesting to overlay a 1km model grid on these to see what we are dealing with when trying to resolve individual fields.

Fig. 2 (Caption): Is this from STATSGO or from individual field samples?

Fig. 3 (Caption): Inferring that heavier precip is more localized?

Fig. 4 (Caption): depths across all sites?

Fig. 5: Hard to see the error bars (busy plot already) - are they important or can they be conveyed in a sentence or two (general trends of increasing w/irrigation amount?).

Fig. 5: They are all underestimating the reported totals, though the slopes are consistent mostly weighted by the very high anchor points (600mm). Very mixed bag at lower values (300mm).

Fig. 6 (Caption): Is this P+I from observations, or output from the schemes?

Fig. 7: What is going on in 2008?

I'm a big disappointed in the analysis/figures. Would have been nice to see some time series of how these schemes are all working over time and in response to precip and precip variability.

To this end, it will be important for LSMs to get the seasonal and sub-seasonal cycle right (including the exact timing of irrigation) if they are to be used for coupled modeling and initialization. So the long-term or annual totals do not tell the whole story.

---

## Referee Comment (RC2) · Anonymous Referee #2 · 30 Sep 2016

I find the study interesting and relevant. A better account of irrigation impact and dynamics in LSM is definitely an area that needs investigation.

I do miss more specific information on the actual linkage between the described irrigation routines and the so-called hyper-resolution LSM. An actual example on this would have been a particularly strong additional element. As a minimum, a more detailed description on the potential integration should be provided along with its feasibility (i.e., input requirements and sources, crop-specific calibrations, limitations etc) for large-scale application. In addition some clarifications to the methodology and findings are

needed as detailed below.

Specific comments:

1) Hyper-resolution needs to be properly defined. For me hyper-resolution intuitively refers to something that is very fine and very well resolved (i.e. at the meter scale) but that is obviously not the case here. 2) L136 – 66: I think that the points made in these sections are valid but I do think that framing would benefit from a slightly more streamlined and ordered structure, if possible. 3) L67-68: How was the critical field scale established? 4) L91: Not sure what is referred to here in terms of the critical LSM scale. 5) L94-95: I would hope you could be a little more specific when talking about the next generation of hyper-resolution LSM and operational weather forecast models; what those this statement imply? 6) L100-102: I would save the specifics of the irrigation routines to the method section. 7) L113: I find Fig. 1 pretty poor and not that informative. As a minimum, you will need a meaningful background image for the field boundary overlay. 8) L117: Why the reference to alfalfa here the entire area in under maize production? 9) L125-130: I think that you need to be more specific on the actual datasets used in this study. I see no description of the meteorological forcing data used. 10) L134: The full names of the irrigation schemes should be given here as well. 11) L135: Why is "(CM)" given here? Same issue with "(H)" in next sentence. The reference/link is not evident from the text. 12) Section 2.2.1: I'm a little confused about the differentiation between CM and HM. HM also seems to be linked to Hydrus but not CM? May need a separate description of HM if that is the case or use CM consistently throughout. 13) L150-151: The inputs (e.g., meteorological data, crop biophysical parameters) to the model are not well described here or in Section 2.1. 14) L195: "was triggered" 15) L208: How was daily ETr determined? 16) L222: HM or CM? See previous comment. 17) L243: So are you saying that you used a non-dynamic (i.e., the same) LAI time-series for all years? Why not consider inter-annual variations in phenology? Does these descriptions of HM also apply to CM? 18) L244 and L250: The sentence "In addition, HM. . .." is repeated here. 19) L307: There's an

issue with the figure numberings. Fig. 5 referred to here is Fig. 6. 20) L317: This is not Fig. 6 but Fig. 5. 21) L317-323: I'm confused about these numbers, which seem somewhat conflicting. It is stated that both CM and PD are near the historical average. But then it is mentioned that CM is 80 mm lower, the same as ET. In addition, the percentages differ. I also find it difficult to verify these numbers based on the figure. These issues will need to be clarified. 22) L323: Fig. 5? 23) Section 3.5: Why is ET and PD not mentioned here? 24) Section 3.6: In Fig. 7, the CM and ET colors can't be distinguished. 25) L353-354: The historically reported yield should also be plotted on the figure for comparison. 26) L371: Was the 30% reduced irrigation need described/mentioned in the results? 27) L401-413: This section is a little hard to follow and should be rewritten for better clarity. 28) Section 4.4: This section is very brief and would benefit from a much more substantial and elaborate description of the feasibility and limitations associated with the integration of the routines in the LSMs. 29) L447: Isn't the 1 km scale often too coarse to resolve field-specific irrigation dynamics?

---

## Short Comment (SC1) · 20 Oct 2016

I find the study interesting and relevant. A better account of irrigation impact and dynamics in LSM is definitely an area that needs investigation. I do miss more specific information on the actual linkage between the described irrigation routines and the so-called hyper-resolution LSM. An actual example on this would have been a particularly strong additional element. As a minimum, a more detailed description on the potential integration should be provided along with its feasibility (i.e., input requirements and sources, crop-specific calibrations, limitations etc) for largescale application. In addition some clarifications to the methodology and findings are needed as detailed below.

*Thank you for the thoughtful review.*

Specific Comments:

1) Hyper-resolution needs to be properly defined. For me hyper-resolution intuitively refers to something that is very fine and very well resolved (i.e. at the meter scale) but that is obviously not the case here.

*We adopted the language from Wood et al. 2011 but will clarify.*

2) L136 – 66: I think that the points made in these sections are valid but I do think that framing would benefit from a slightly more streamlined and ordered structure, if possible.

*Thank you for the suggestion. We will rework the section and streamline where possible.*

3) L67-68: How was the critical field scale established?

*This is the scale at which human-water decisions are made at due to the history of land partitioning. The inherent geometry is dictated in this landscape. We will clarify in the text.*

4) L91: Not sure what is referred to here in terms of the critical LSM scale.

*See comment above.*

5) L94-95: I would hope you could be a little more specific when talking about the next generation of hyper-resolution LSM and operational weather forecast models; what those this statement imply?

*We will try and clarify in revision.*

6) L100-102: I would save the specifics of the irrigation routines to the method section.

*We felt a brief description was helpful to introduce the overall framework of the paper.*

7) L113: I find Fig. 1 pretty poor and not that informative. As a minimum, you will need a meaningful background image for the field boundary overlay.

*Thank you for the suggestion. We will update with a 1 km grid along with a more meaningful background image.*

8) L117: Why the reference to alfalfa here the entire area in under maize production?

*Agreed, will update with referenced alfalfa here.*

9) L125-130: I think that you need to be more specific on the actual datasets used in this study. I see no description of the meteorological forcing data used.

*We will update with a description of the meteorological forcing data. HESS now requires a data availability section we will include.*

10) L134: The full names of the irrigation schemes should be given here as well.

*We will add this in.*

11) L135: Why is "(CM)" given here? Same issue with "(H)" in next sentence. The reference/link is not evident from the text.

*This is the abbreviation for the irrigation routine.*

12) Section 2.2.1: I'm a little confused about the differentiation between CM and HM. HM also seems to be linked to Hydrus but not CM? May need a separate description of HM if that is the case or use CM consistently throughout.

*CM and HM are linked. Hydrus uses the outputs from HM.*

13) L150-151: The inputs (e.g., meteorological data, crop biophysical parameters) to the model are not well described here or in Section 2.1.

*We will update with a description of the meteorological forcing data. Additionally, a crop coefficient table will be added. HESS now requires a data availability section we will include.*

14) L195: "was triggered"

*We will update.*

15) L208: How was daily ETr determined?

*From the meteorological dataset.*

16) L222: HM or CM? See previous comment.

*See previous comment.*

17) L243: So are you saying that you used a nondynamic (i.e., the same) LAI time-series for all years? Why not consider inter-annual variations in phenology? Does these descriptions of HM also apply to CM?

*No, just a single LAI time series for all irrigation routines. The LAI time series is on the daily time step and varies from year-to-year. The description will be updated for clarity.*

18) L244 and L250: The sentence "In addition, HM...." is repeated here.

*Will update and remove the repetition.*

19) L307: There's an issue with the figure numberings. Fig. 5 referred to here is Fig. 6.

*Yes, will update the figure number.*

20) L317: This is not Fig. 6 but Fig. 5.

*Yes, will update the figure number.*

21) L317-323: I'm confused about these numbers, which seem somewhat conflicting. It is stated that both CM and PD are near the historical average. But then it is mentioned that CM is 80 mm lower, the same as ET. In addition, the percentages differ. I also find it difficult to verify these numbers based on the figure. These issues will need to be clarified.

*Agreed, this does need clarification. The slopes are similar but with an offset. The percentages will also be clarified.*

22) L323: Fig. 5?

*Yes, will update the figure number.*

23) Section 3.5: Why is ET and PD not mentioned here?

*The don't have a soil consideration within the routine and so soil texture will not have an impact on their numbers. This will be mentioned in the text.*

24) Section 3.6: In Fig. 7, the CM and ET colors can't be distinguished.

*We will update both colors and line weights for clarity.*

25) L353-354: The historically reported yield should also be plotted on the figure for comparison.

*We only have historical yield for years prior to the study.*

26) L371: Was the 30% reduced irrigation need described/mentioned in the results?

*We will try and clarify in revision.*

27) L401-413: This section is a little hard to follow and should be rewritten for better clarity.

*Agreed, this will be reworked for clarity.*

28) Section 4.4: This section is very brief and would benefit from a much more substantial and elaborate description of the feasibility and limitations associated with the integration of the routines in the LSMs.

*We will expand section in revision as suggested.*

29) L447: Isn't the 1 km scale often too coarse to resolve field-specific irrigation dynamics?

*Not necessarily for this landscape. The land is partitioned into 0.8 km sections. Often irrigation decisions are made for uniform conditions. Some sub field decisions using precision agriculture are now available but not widely used yet.*

---

## Author Comment (AC1) · 20 Oct 2016

Overall Comments:

Overall, this is a well-written manuscript describing 4 new ways to account for irrigation that could be used by managers and modelers alike. This type of work is much needed, as the human element/drivers of new LSM physics remain a challenge in how to account for them and prescribe them accurately. This is also a novel dataset put to good use. The schemes use sound assumptions and represent an array of complexities. The paper is a worthwhile contribution, but becomes a bit thin in the results section and a few of the major limitations are glossed over and require further discussion.  As a result, I recommend major revisions in order to help the manuscript become more impactful and useful for irrigation-related studies. In addition, I strongly recommend that, if possible, the results/analysis be extended to time series and sub-annual breakdowns of irrigation water vs.   precipitation (and variability) for each of these schemes.  Much of the utility for managers and more so for modelers will be on the diurnal and sub-seasonal scales, in which they need to obtain the water balance,  soil moisture,  and fluxes correct in order to couple to the atmosphere and represent the precipitation connection more accurately (i.e coupling). Also missing is the broader applicability of these schemes outside of this unique, well-instrumented and reported-on field/domain. Other locations with less decision-making data points or coarser precipitation will no doubt find greater challenges.

*Thank you for the thoughtful comments. We will do our best to address your main concerns.*

Specific Comments:

L24: What is difference between a conservative and water savings routine? Sounds similar if you do not know the terminology. This is explained better in the paper itself, but maybe a word or two in the abstract could help better clarify what is meant by each.

*Thank you for the suggestion. We will update the abstract to be more consistent with the manuscript.*

L29: Is the actual transition of information and decision making part of this paper? Or is it suggested that it would be valuable in the future for managers? If the latter (which according to the paper itself there is no transition or decision making taking place (yet!), then please clarify this here to suggest it may be useful in the future (not that it already has been useful).

*Agreed, we will update to make more explicit that it could be useful in the future.*

L52: might want to mention that the impact of SM on these is really modulated by the flux contribution to the atmosphere (SHF, LHF, or evap fraction, or just ET). So getting the SM-Flux relationship correct is critical, and i.e irrigation is essential as a component of that.

*Indeed, the presence of irrigation doesn't necessarily impact the flux rates – we will update to include the SM-Flux relationship.*

L58: Which are the 'both' here? L59: Is there a predictive nature to irrigation decisionmaking? Do Calendars vs. Consultants vs. Probe percentages change over time due to other factors

(technology, financial, drought, etc.)? Are consultant-based decisions consistent (is the advice consistent) over time?

*Both the risk-aversion side of decision making and from biophysical requirements. Gibson, 2015 identified that the majority of irrigated fields were irrigated approximately 50mm more that crop water demand.*

*Gibson, J.P., Estimation of Deep Drainage Differences between Till and No-Till Irrigated Agriculture. Master's Thesis, 2015.*

L99: Not clear what is meant by 'irrigation triggering regimes'? Earlier (abstract) they were referred to as 'routines' that could be incorporated into LSMs. Regimes suggest something different?

*We will update to keep routine consistent throughout the manuscript.*

L122: What is the native resolution of SSURGO relative to the study area and field scale?

*Greater than field scale but still does well for in- field observations.*

L125: Same for SPNRD.

*Data is on the field scale, total volume pumped for irrigated area.*

Section 2.2: Based on the descriptions of these, are this ranging from the most simple to most complex (in order)?

*Yes, simplest to most complex. We will clarify.*

For H, would it be possible that the minimal yield loss could be set so high as to represent larger irrigation than in CM?

*Not possible within the constraints of irrigation depths and frequency (3 days for the lateral to move 360 degrees). The CM is triggered with no constraint on irrigation frequency.*

L168: 'amount of water'

*Will update to target total of irrigation plus in-season rainfall.*

L179: Has this approach been used in the past? There are no references, and based on interviews and expert knowledge. How did you come up with 6.5 exactly? If the ultimate goal is to have this in an LSM, I can envision that it might be very sensitive to this 6.5 number and thus overly simplistic. Are there any other knobs to turn?

*The SPNRD recommends a total amount of P+I of 650 mm within the growing season. In other areas this could be informed by growing season ET totals. The irrigation season is approximately 100 days long based on typical irrigation patterns. So 650mm/100 days is 6.5 mm/day in the absence of rainfall to meet this demand. The work of Sharma and Irmak 2012 quantify net irrigation requirement around NE.*

*Sharma, V. and Irmak, S.: Mapping spatially interpolated precipitation, reference evapotranspiration, actual crop evapotranspiration, and net irrigation requirements in Nebraska: Part II Actual evapotranspiration and net irrigation requirements, Trans. ASABE (American Soc. Agric. Biol. Eng., 55(3), 923–936, doi:10.13031/2013.41524, 2012.*

L185: This sounds reasonable as first order approximations for extreme rainfall. What about the low-intermediate rainfall conditions and the speed of drainage? Should the delay estimates be constant regardless of the soil type (conductivity), land cover, and precipitation rate?

*Low rainfall rates (<6.5 mm/day) will not lead to a delay in irrigation application and this is consistent with discussions with producers in the area. Significant drainage is not expected within the growing season due to ET demand. Highly conductivity soils would require a shorter delay, however maize is not typically produced in such soils. Land cover will change but these algorithms are specific to maize.*

L243: What is meant by seasonal dynamic?

*Will update to daily time series.*

L280: All assumptions embedded in these approaches have been explained and seem reasonable. The proof is in the pudding, of course, and the results will bear that out. However, it might be useful to summarize what the input requirements and the assumed/tunable parameters are for each approach as well, if they are to be used in LSMs. An example here is the date ranges that are used. 6.5 is another as is -500cm, and the depths of the soil pressure.

*Yes, a summary table will be added to explain how these parameters would need to be update based off different management practices.*

L284: Where are they located with respect to the study site and the fields? Should some kind of interpolation (or average) be used as well?

*The average of the 7 gauges was used.*

L293: Mean ETc?

*Will update to the range between the highest and lowest rainfall totals.*

L294: Totals of what?

*Will update to precipitation totals.*

L297: This is critical. The 4 schemes rely on P as the most important input (right?). Forcing for LSMs comes from satellite and gauge-based datasets, likely much coarser (e.g. .125-deg) than the <1km field scale. How will this be addressed? How can we capture the irrigation variability without knowing that of Precip?

*Indeed, P is the most important input in both the routines and in the field. Decision making occurs from both radar estimations and in-field gauge readings. On shorter timescales (day to weekly), rainfall variability tends to be large. However, on the monthly to seasonal scale, variability tends to decrease. This is in part why we have focused on seasonal totals.*

L325: I think a lot more could be said - this is the critical result/figure from this paper. There is a lot of error bar info on there and other aspects that could be discussed. The low bias stands out and is significant.

*Yes, the low bias motivates the recommendation of 50-75 mm reduction in irrigation application. The fact that irrigation application is in excess of crop water demand is in line with Gibson, 2015.*

L328: 'Regimes' again.

*Will update to routine.*

L338: See earlier comment. This is a major limitation to all of these approaches and modeling irrigation at this scale.

*See comment above (L325).*

L356: What does this imply about the assumed yield-irrigated amount relationship? That they underestimate and still didn't impact yield is even more surprising. There must be a lot of leeway (i.e. overwatering?).

*This is the motivation and focus of the ongoing cost-share program funded by Coca-Cola within the study area. This will be the focus of Gibson's PhD looking at corporate supply chain sustainability and scientifically sound water savings numbers. More to come over the next few years.*

L373: You are saying that, based on these models, you can get away with much less water and still produce the same yield, correct? Isn't that something that should have been quantified in the past (or known by the farmers)? Or is this still largely unknown?
How certain are we that the models are correct and that the yield will still be met?

*See comment above.*

L384: Supports the need for a bit further analysis/figures looking into the time series of the results.

*We will investigate this.*

L392: This was alluded to in an earlier comment: How can we know that prior decision making holds in the future or during other conditions not in the recent historical record?

*We can only hypothesize about future conditions, continued monitoring of irrigation application will be important with the continued trend of irrigation technology adoption.*

L404: Why? Is it because soil types here are so are similar, with slowly varying properties?

*Measurement of the soil properties is currently in progress. This was a surprising result indeed!*

L433: How about a controlled experiment/field to test sensitivity and realism of these schemes and resultant quantities? Is that reasonable in the future?

*Integration of these considerations within a producer's operation may be feasible and is indeed the focus of current work. However, the suggestion of a producer strictly following these mechanistic routines and abandoning their own "know-how" is unlikely to be well received other than at research and extension centers with more control. Producers are unlikely to make decisions that will affect their economics. Perhaps a program where we compensate the producer for yield losses could be implemented in the future.*

L443: Any predictive capabilities?

*We will investigate this.*

Section 4: This discussion section was welcome - lot of areas that need study but this is a good start.

L453: 'may be useful'?

*Will update.*

Section 5: The conclusions are a bit thin, and perhaps should focus on some of the limiting factors and broader/future applicability (precip forcing, decision making, soil properties).

*Thank you for the suggestion.*

Fig. 1: Hard to tell exactly where these fields are as this box points to a point on the corner of CO and NE.

Fig. 1: Might be interesting to overlay a 1km model grid on these to see what we are dealing with when trying to resolve individual fields.

*Agreed, will take a look and see if this helps.*

Fig. 2 (Caption): Is this from STATSGO or from individual field samples?

*SSURGO data downloaded from web soil survey and parsed via the NRCS toolkit.*

Fig. 3 (Caption): Inferring that heavier precip is more localized?

*Thank you for the suggestion.*

Fig. 4 (Caption): depths across all sites?

*Thank you for the suggestion.*

Fig. 5: Hard to see the error bars (busy plot already) - are they important or can they be conveyed in a sentence or two (general trends of increasing w/irrigation amount?).

*Agreed, will surmise in text.*

Fig. 5: They are all underestimating the reported totals, though the slopes are consistent mostly weighted by the very high anchor points (600mm). Very mixed bag at lower values (300mm).

*Thank you for the suggestion.*

Fig. 6 (Caption): Is this P+I from observations, or output from the schemes?

*From observation, will update.*

Fig. 7: What is going on in 2008?

*We will investigate this.*

I'm a big disappointed in the analysis/figures. Would have been nice to see some time series of how these schemes are all working over time and in response to precip and precip variability.

To this end, it will be important for LSMs to get the seasonal and sub-seasonal cycle right (including the exact timing of irrigation) if they are to be used for coupled modeling and initialization. So the long-term or annual totals do not tell the whole story.

*We will investigate this but are somewhat limited by the data only being at annual totals. We are working on a followup paper using energy use as a proxy to estimate subdaily irrigation rates in the area.*

---

## Author Response (AR1)

Dear Prof. McCabe,

We would like to thank you and the two reviewers for your time and excellent comments regarding our manuscript, titled "A case study of field-scale maize irrigation patterns in Western Nebraska: Implications to water managers and recommendations for hyper-resolution land surface modelling". After careful analysis of all the comments, we have made extensive revisions to our manuscript. You can find our detailed responses to the reviewers' comments (shown in red italics) and the changes we made to the manuscript in the following sections. We have also included a marked up version of the original manuscript.

On the behalf of all coauthors, I hope that this revised version would meet the publication standard of Hydrology and Earth System Sciences (HESS) and inclusion in the Eric F. Wood special issue. Please let us know if there are more questions and comments about the manuscript.

Sincerely,

Justin Gibson

School of Natural Resources

University of Nebraska-Lincoln, USA

Comments to the Author:
Dear Justin.

Thank you for your manuscript, submitted to HESS as part of the Special Issue on "Observations and modeling of land surface water and energy exchanges across scales". After reviewing your contribution and the comments from the two referees, I am requesting that you provide some additional details and revisions so that I can further assess it for publication. I am optimistic that these revisions should be fairly straightforward to implement, as they are mostly structural or require additional paragraphs for analysis or interpretation. I have detailed some of the more critical suggestions below, but draw your attention to the detailed comments provided by each of the referees in their respective reports. I encourage you to carefully consider these in your revised version, providing details on where the manuscript has been updated in your response.

1. In line with RC1, I agree that a time series/sub-annual breakdowns would be very useful, but also appreciate the data limitations. Refined temporal analysis would be especially important for active management of irrigation systems, as well as for the LSM community in better representing such systems in modeling approaches. Where possible, try to expand upon the temporal aspect, either explicitly in the results or more generally in the discussion sections.

*Section 3.7 was added to discuss the sub-seasonal irrigation time series. Figure 8 presents the results.*

**L376-382: 3.7 Simulated Growing Season Irrigation Application**
*Daily time series of simulated irrigation application can be seen in Fig. 8. Data for observed sub-growing season irrigation application is unavailable. Irrigation application tends to begin later in the growing season for the two routines that consider soil (CM and H). This is likely due to the routines first allowing soil moisture to be depleted before irrigation is triggered. The amount of soil moisture storage is typically near field capacity but in exceptionally dry years (2012) this storage is reduced and thus will lead to less of a delay.*

2. Both reviewers identify the need for some discussion on the broader application of these schemes, both in terms of their generality beyond this specific location, as well as their potential integration into "hyper-resolution" type schemes. Section 4.4 is an obvious area where these concepts could be expanded upon. I understand that an actual example may not be feasible, but you can certainly identify some of the challenges and opportunities that such a scheme may present.

*We have expanded this section as requested. We also note 2 followup papers are in preparation that explore this topic (Gibson et al. 2017, Ag water management, and Lawston et al. 2017 HESS)*

*L461: The four irrigation routines although biased, capture year-to-year variation in irrigation in Western Nebraska. Given the widespread use of center-pivots we expect the irrigation routines*

*to be appropriate for the HPA and into parts of the eastern USA. Gibson (2016) provides a fuller assessment of irrigation behavior throughout central Nebraska. We note that it is unclear how these routines would behave in areas with center-pivot outside the USA (i.e. Brazil, South Africa, Australia), where energy costs for pumping may be more restricting and drive human-decisions on irrigation. Assessment of these routines in those areas would require further validation.*

*We believe the routines combined with a reasonable bias correction could be easily incorporated into future hyper-resolution LSMs with the above routine descriptions and readily available LSM model output or datasets (see Table 1). Clearly accurate and local precipitation is critical in driving these irrigation routines and capturing producer behavior. This topic deserves more research, particularly and the opportunity to combine low cost in-situ gages with radar and remote sensing products. Additionally, we note the four routines could be run offline in order to provide reasonable guesses of applied irrigation for a given irrigation season. This may be beneficial in representing processes not explicitly considered in LSMs (Kumar et al. 2015), or making future assessments and recommendations about water availability for managers. Finally, the four routines provide reasonable irrigation bounds and more importantly predictions about decreases in irrigation as technology is introduced and adopted in novel areas.*

3. An additional section (or combined within Section 4.4) focusing on possible implementation requirements or issues may also be useful: this may go some way to addressing both referee comments on the feasibility of the approach for broader scale application.

*Table 1 is provided that summarizes the inputs and tunable parameters needed for each routine. In addition Lawston 2017 HESS explores the role of irrigation physics in the NOAH LSM.*

4. In addition to summarizing key results, the conclusions can synthesize some of these discussions to provide a broader scale context for the work.

*The conclusions were modified to include the key future direction we recommend for understanding irrigation behavior in this area and the HPA in general. That is providing realtime local precipitation to producers. This is a hot topic being actively pursued by private industry as well. The ability to merge low cost sensor networks with radar and satellite products would be a huge benefit to producers and water managers alike.*

*L482: In this work we describe four plausible and relatively simple irrigation routines that could be coupled to the next generation of hyper-resolution LSMs operating at scales of 1 km or less. The crop model irrigation outputs reproduce the year-to-year variability of the observed irrigation amounts with a low bias of 80 mm yr-1.  Predictions from the vadose zone model indicate potential irrigation savings of up to 120 mm yr-1 for maize. In addition, daily precipitation variability across the study area was found to introduce significant variability in*

*daily irrigation decision making depending on which value was considered. Future work could focus on providing accurate realtime 1 km daily precipitation products through a combination of in-situ low cost gages, radar, and satellite remote sensing. Accurate and realtime precipitation remains a critical weakness in these rural and vast landscapes. Given the clustering of irrigation fields in Western Nebraska, the number of in-situ gages needed could be significantly reduced to provide high density networks in key areas. Findings from the work may be useful to local water managers and stakeholders in evaluating potential water saving technologies. In addition, the simple routines could be coupled to future hyper-resolution land surface models that seek to understand the degree of land surface atmospheric coupling and consequences to operational forecasts. This understanding is essential as society continually recognizes the importance of human activities on the global water cycle and invests more resources to understand the water-food-energy nexus.*

5. Carefully review language and grammar: I'm not sure you mean "predications" on Line 449?

*Thank you we have made the change.*

Overall, I believe that the manuscript will benefit from a focused revision, addressing these and the specific referee comments: I look forward to receiving an updated version.

Best wishes,
Matt

Reviewer 1:

Overall Comments:

Overall, this is a well-written manuscript describing 4 new ways to account for irrigation that could be used by managers and modelers alike. This type of work is much needed, as the human element/drivers of new LSM physics remain a challenge in how to account for them and prescribe them accurately. This is also a novel dataset put to good use. The schemes use sound assumptions and represent an array of complexities. The paper is a worthwhile contribution, but becomes a bit thin in the results section and a few of the major limitations are glossed over and require further discussion.  As a result, I recommend major revisions in order to help the manuscript become more impactful and useful for irrigation-related studies. In addition, I strongly recommend that, if possible, the results/analysis be extended to time series and sub-annual breakdowns of irrigation water vs.   precipitation (and variability) for each of these schemes.  Much of the utility for managers and more so for modelers will be on the diurnal and sub-seasonal scales, in which they need to obtain the water balance, soil moisture,  and fluxes correct in order to couple to the atmosphere and represent the precipitation connection more accurately (i.e coupling). Also missing is the broader applicability of these schemes outside of this unique, well-instrumented and reported-on field/domain. Other locations with less decision-making data points or coarser precipitation will no doubt find greater challenges.

*Thank you for the thoughtful comments. We will do our best to address your main concerns. We have added a section (3.7) of daily dynamics of each of the irrigation schemes in a wet and dry year. It is clear the water savings are due to starting irrigation later in the season by better harvesting available soil water storage. Without monitoring or modeling this the producer is left with a tough decision on when best to irrigate. In addition, we are working with a cost sharing program to bring useful technologies to this area and is the focus of J. Gibson's PhD. Initial discussions with water managers and producers indicate a real desire to increase monitoring (rainfall in particular) with realtime decisions through pivot telemetry. This work will serve as a key study to continue to build these relationships and make lastly changes in the real world to conserve water and sustain critical livelihoods.*

Specific Comments:

L24: What is difference between a conservative and water savings routine? Sounds similar if you do not know the terminology. This is explained better in the paper itself, but maybe a word or two in the abstract could help better clarify what is meant by each.

*Updated abstract to:*

*L24: Here we find the most yield-conservative irrigation routine (crop model).*

L29: Is the actual transition of information and decision making part of this paper? Or is it suggested that it would be valuable in the future for managers? If the latter (which according to the paper itself there is no transition or decision making taking place (yet!), then please clarify this here to suggest it may be useful in the future (not that it already has been useful).

*Agreed, we will update to make more explicit that it could be useful in the future. Text updated to:*

*L29: The results from the various irrigation routines and associated yield penalties will be valuable for future consideration by local water managers to be informed by the potential value of water savings technologies and irrigation practices.*

L52: might want to mention that the impact of SM on these is really modulated by the flux contribution to the atmosphere (SHF, LHF, or evap fraction, or just ET). So getting the SM-Flux relationship correct is critical, and i.e irrigation is essential as a component of that.

*Indeed, the presence of irrigation doesn't necessarily impact the flux rates – we will update to include the SM-Flux relationship. Update text to:*

*L52: Although the presence of irrigation doesn't necessarily impact soil moisture contribution to the atmosphere, the soil moisture-flux relationship is critical to surface energy balance and atmospheric dynamics.*

L58: Which are the 'both' here?

*Both are affected by both. Changed sentence for clarity.*

*Both the risk-aversion side of decision making and from biophysical requirements. Gibson, 2015 identified that the majority of irrigated fields were irrigated approximately 50mm more that crop water demand.*

*Gibson, K.E.B: More Crop per Drop: Benchmarking On-Farm Irrigation Water Use for Crop Production. Master's Thesis, 2016.*

*L58: More complicating is that both irrigation timing and volumes are based on human decision making processes and biophysical requirements (Gibson, 2016).*

L59: Is there a predictive nature to irrigation decision making? Do Calendars vs. Consultants vs. Probe percentages change over time due to other factors (technology, financial, drought, etc.)? Are consultant-based decisions consistent (is the advice consistent) over time?

*K. Gibson (2016) found only 45% of irrigation volumes can be explained by biophysical factors. The remaining variation is likely due to human decision making. Seems to be a challenging social ecological system to understand, particularly for prediction.*

L99: Not clear what is meant by 'irrigation triggering regimes'? Earlier (abstract) they were referred to as 'routines' that could be incorporated into LSMs. Regimes suggest something different?

*We will update to keep routine consistent throughout the manuscript.*

*Replaced all "regimes" with "routine"*

L122: What is the native resolution of SSURGO relative to the study area and field scale?

*Greater than field scale but still does well for in- field observations.*

L125: Same for SPNRD.

*Data is on the field scale, total volume pumped for irrigated area.*

Section 2.2: Based on the descriptions of these, are this ranging from the most simple to most complex (in order)?

*Yes, simplest to most complex. We will clarify.*

*L144: Moreover, the four routines can be easily coupled or implemented into LSMs where PD is the simplest routine, and H the most complex.*

For H, would it be possible that the minimal yield loss could be set so high as to represent larger irrigation than in CM?

*Not possible within the constraints of irrigation depths and frequency (3 days for the lateral to move 360 degrees). The CM is triggered with no constraint on irrigation frequency.*

L168: 'amount of water'

*Updated to clarify the depth of water.*

*L166-168: Irrigation depth is determined by the deficit of soil moisture defined by the current moisture level subtracted from 95% of field capacity.*

L179: Has this approach been used in the past? There are no references, and based on interviews and expert knowledge. How did you come up with 6.5 exactly? If the ultimate goal is to have this in an LSM, I can envision that it might be very sensitive to this 6.5 number and thus overly simplistic. Are there any other knobs to turn?

*The SPNRD recommends a total amount of P+I of 650 mm within the growing season. In other areas this could be informed by growing season ET totals. The irrigation season is approximately 100 days long based on typical irrigation patterns. So 650mm/100 days is 6.5 mm/day in the absence of rainfall to meet this demand. The work of Sharma and Irmak 2012 quantify net irrigation requirement around NE. This same type of procedure could be extended across the HPA to determine daily irrigation intensity.*

*Sharma, V. and Irmak, S.: Mapping spatially interpolated precipitation, reference evapotranspiration, actual crop evapotranspiration, and net irrigation requirements in Nebraska: Part II Actual evapotranspiration and net irrigation requirements, Trans. ASABE (American Soc. Agric. Biol. Eng., 55(3), 923–936, doi:10.13031/2013.41524, 2012.*

L185: This sounds reasonable as first order approximations for extreme rainfall. What about the low-intermediate rainfall conditions and the speed of drainage? Should the delay estimates be constant regardless of the soil type (conductivity), land cover, and precipitation rate?

*Low rainfall rates (<6.5 mm/day) will not lead to a delay in irrigation application and this is consistent with discussions with producers in the area. Significant drainage is not expected within the growing season due to ET demand. Highly conductive soils would require a shorter delay, however maize is not typically produced in such soils. Land cover will change but these algorithms are specific to maize.*

L243: What is meant by seasonal dynamic?

*Updated to daily time series.*

*L255-256: For each year's growing season we simulated a daily LAI time series using HM.*

L280: All assumptions embedded in these approaches have been explained and seem reasonable. The proof is in the pudding, of course, and the results will bear that out. However, it might be useful to summarize what the input requirements and the assumed/tunable parameters are for each approach as well, if they are to be used in LSMs. An example here is the date ranges that are used. 6.5 is another as is -500cm, and the depths of the soil pressure.

*Yes, Table 1 provided a summary of key inputs and tunable parameters for each routine.*

L284: Where are they located with respect to the study site and the fields? Should some kind of interpolation (or average) be used as well?

*The average of the 7 gauges was used. It is clear that local rainfall data is needed for optimal irrigation management and a focus of future work.*

L293: Mean ETc?

*We had only 1 ETc estimate.*

L294: Totals of what?

*Will update to precipitation totals.*

*L305-306: In general, as precipitation totals increased, the range of seasonal precipitation totals observed by the 7 gauges increased as well (slope = 0.246 mm $yr^{-1}$, $R^2$ = 0.38).*

L297: This is critical. The 4 schemes rely on P as the most important input (right?). Forcing for LSMs comes from satellite and gauge-based datasets, likely much coarser (e.g. .125-deg) than the <1km field scale. How will this be addressed? How can we capture the irrigation variability without knowing that of Precip?

*Indeed, P is the most important input in both the routines and in the field. Decision making occurs from both radar estimations and in-field gauge readings. On shorter timescales (day to weekly), rainfall variability tends to be large. However, on the monthly to seasonal scale, variability tends to decrease. This is in part why we have focused on seasonal totals. Future work with this area will include the development and installation of a low cost met. station network delivered in realtime to producers. We added reference to Gibson 2016 which tackles this issue in more depth in central Neb.*

L325: I think a lot more could be said - this is the critical result/figure from this paper. There is a lot of error bar info on there and other aspects that could be discussed. The low bias stands out and is significant.

*Yes, the low bias motivates the recommendation of 50-75 mm reduction in irrigation application. The fact that irrigation application is in excess of crop water demand is in line with Gibson, 2015.*

L328: 'Regimes' again.

*Will update to routine.*

L338: See earlier comment. This is a major limitation to all of these approaches and modeling irrigation at this scale.

*See comment above (L325). Indeed we need the local P data. We hope the new NASA GPM 4 km product will be useful here.*

L356: What does this imply about the assumed yield-irrigated amount relationship? That they underestimate and still didn't impact yield is even more surprising. There must be a lot of leeway (i.e. overwatering?).

*This is the motivation and focus of the ongoing cost-share program funded by Coca-Cola within the study area. This will be the focus of Gibson's PhD looking at corporate supply chain*

*sustainability and scientifically sound water savings numbers. More to come over the next few years.*

L373: You are saying that, based on these models, you can get away with much less water and still produce the same yield, correct? Isn't that something that should have been quantified in the past (or known by the farmers)? Or is this still largely unknown?
How certain are we that the models are correct and that the yield will still be met?

*See comment above (L356).*

L384: Supports the need for a bit further analysis/figures looking into the time series of the results.

*We have added a section of the daily time series of each model (Figure 8 and section 3.7).*

L392: This was alluded to in an earlier comment: How can we know that prior decision making holds in the future or during other conditions not in the recent historical record?

*We can only hypothesize about future conditions. Continued monitoring of irrigation application will be important with the continued trend of irrigation technology adoption.*

L404: Why? Is it because soil types here are so are similar, with slowly varying properties?

*Measurement of the soil properties is currently in progress. This was a surprising result indeed! Gibson 2016 also explored this and found soils had minimal impact on explaining irrigation amounts. Seems about 30% of irrigation variability in central NE can be explained by available water holding capacity. More to come on this for Gibson (2016 in review) and work with this project.*

L433: How about a controlled experiment/field to test sensitivity and realism of these schemes and resultant quantities? Is that reasonable in the future?

*Integration of these considerations within a producer's operation may be feasible and is indeed the focus of current work. However, the suggestion of a producer strictly following these mechanistic routines and abandoning their own "know-how" is unlikely to be well received other than at research and extension centers with more control. Producers are unlikely to make decisions that will affect their economics. Perhaps a program where we compensate the producer for yield losses could be implemented in the future. Some existing literature on this is from the Nebraska Water Balance Alliance.*

L443: Any predictive capabilities?

*Not sure.*

Section 4: This discussion section was welcome - lot of areas that need study but this is a good start.

L453: 'may be useful'?

*Updated.*

Section 5: The conclusions are a bit thin, and perhaps should focus on some of the limiting factors and broader/future applicability (precip forcing, decision making, soil properties).

*We added a few sentences about providing better rainfall products, which is the lowest hanging fruit in our opinion.*

*L477: Future work could focus on providing accurate realtime 1 km daily precipitation products through a combination of in-situ low cost gages, radar, and satellite remote sensing. Accurate and realtime precipitation remains a critical weakness in these rural and vast landscapes. Given the clustering of irrigation fields in Western Nebraska, the number of in-situ gages needed could be significantly reduced to provide high density networks in key areas.*

Fig. 1: Hard to tell exactly where these fields are as this box points to a point on the corner of CO and NE.

Fig. 1: Might be interesting to overlay a 1km model grid on these to see what we are dealing with when trying to resolve individual fields.

*We have added a 1 km grid to the figure.*

Fig. 2 (Caption): Is this from STATSGO or from individual field samples?

*SSURGO data downloaded from web soil survey and parsed via the NRCS toolkit. Changed caption.*

Fig. 3 (Caption): Inferring that heavier precip is more localized?

*Thank you for the suggestion.*

Fig. 4 (Caption): depths across all sites?

*Changed caption to all sites.*

Fig. 5: Hard to see the error bars (busy plot already) - are they important or can they be conveyed in a sentence or two (general trends of increasing w/irrigation amount?).

*After consideration we left error bars on plot for completeness of illustrating the mean and its uncertainty. Felt visual was stronger message than describing values in text.*

Fig. 5: They are all underestimating the reported totals, though the slopes are consistent mostly weighted by the very high anchor points (600mm). Very mixed bag at lower values (300mm).

*Yes, clearly risk aversion behavior compared to modeled needs.*

Fig. 6 (Caption): Is this P+I from observations, or output from the schemes?

*From observations, updated caption.*

Fig. 7: What is going on in 2008?

*Not totally sure, perhaps forcing data was off?*

I'm a big disappointed in the analysis/figures. Would have been nice to see some time series of how these schemes are all working over time and in response to precip and precip variability.

To this end, it will be important for LSMs to get the seasonal and sub-seasonal cycle right (including the exact timing of irrigation) if they are to be used for coupled modeling and initialization. So the long-term or annual totals do not tell the whole story.

*We will investigate this but are somewhat limited by the data only being at annual totals. We are working on a followup paper using energy use as a proxy to estimate subdaily irrigation rates in the area.*

*Section 3.7 was added to discuss the sub-seasonal irrigation time series. Figure 8 presents the results.*

***L373-379: 3.7 Simulated Growing Season Irrigation Application***
*Daily time series of simulated irrigation application can be seen in Fig. 8. Data for observed sub-growing season irrigation application is unavailable. Irrigation application tends to begin later in the growing season for the two routines that consider soil (CM and H). This is likely due to the routines first allowing soil moisture to be depleted before irrigation is triggered. The amount of soil moisture storage is typically near field capacity but in exceptionally dry years (2012) this storage is reduced and thus will lead to less of a delay.*

[Figure]

*Fig. 8: Simulated growing season cumulative P and P+I with daily P values plotted on secondary y-axis for the 4 irrigation routines. Irrigation starts later for routines that track soil moisture.*

Reviewer 2:

I find the study interesting and relevant. A better account of irrigation impact and dynamics in LSM is definitely an area that needs investigation. I do miss more specific information on the actual linkage between the described irrigation routines and the so-called hyper-resolution LSM. An actual example on this would have been a particularly strong additional element. As a minimum, a more detailed description on the potential integration should be provided along with its feasibility (i.e., input requirements and sources, crop-specific calibrations, limitations etc) for largescale application. In addition some clarifications to the methodology and findings are needed as detailed below.

*Thank you for the thoughtful review. An example LSM routine (NOAH) is currently the focus of a paper in preparation by P. Lawston that should be submitted to HESS by the end of the year. We refer the reviewer to that study. We have added Table 1, which summarizes the needed inputs for each irrigation routine. Given access to those data we anticipate this scheme would be reasonable for the HPA and eastern USA where center-pivots are in operation.*

Specific Comments:

1) Hyper-resolution needs to be properly defined. For me hyper-resolution intuitively refers to something that is very fine and very well resolved (i.e. at the meter scale) but that is obviously not the case here.

*For better of worse, we adopted the language from Wood et al. 2011.*

2) L136 – 66: I think that the points made in these sections are valid but I do think that framing would benefit from a slightly more streamlined and ordered structure, if possible.

*Thank you for the suggestion. We have made some alterations. In addition we added larger context of the importance of irrigation to global food production: L101*

*L101: We note that irrigation is a key component of global food security, accounting for ~40% of global food production and ~20% of all arable land (Molden, 2007; Schultz et al., 2005). No doubt irrigation will continue to expand in the future.*

3) L67-68: How was the critical field scale established?

*We updated the text to:*

*L71-72: This critical scale is defined as where human-water decisions are made at due to the history of land partitioning and the inherent geometry is dictated by this landscape.*

4) L91: Not sure what is referred to here in terms of the critical LSM scale.

*See comment above.*

5) L94-95: I would hope you could be a little more specific when talking about the next generation of hyper-resolution LSM and operational weather forecast models; what those this statement imply?

*I guess it is simply the inclusion of better irrigation physics in LSM schemes and coupling to atmospheric models like NLDAS. We added the citation to Kumar et al. 2015.*

6) L100-102: I would save the specifics of the irrigation routines to the method section.

*We felt a brief description was helpful to introduce the overall framework of the paper.*

7) L113: I find Fig. 1 pretty poor and not that informative. As a minimum, you will need a meaningful background image for the field boundary overlay.

*Thank you for the suggestion. We updated with a 1 km grid along with a more meaningful background image.*

8) L117: Why the reference to alfalfa here the entire area in under maize production?

*Updated to maize referenced ET*

*L124-125: The study area is semi-arid where annual crop referenced (maize) evapotranspiration ($ET_c$) is significantly higher than precipitation (P) (HPRCC, 2016). The 7-year (2008-2014) average annual P is 440 mm/yr and average annual $ET_c$ is 820 (mm/yr), as measured by the High Plains Regional Climate Center weather station (HPRCC, 2016) located within 10 km of the study area near Brule, NE.*

9) L125-130: I think that you need to be more specific on the actual datasets used in this study. I see no description of the meteorological forcing data used.

*We update the section with a description of the meteorological forcing data. HESS now requires a data availability section we included.*

10) L134: The full names of the irrigation schemes should be given here as well.

*Updated the text to:*

*L140-142: In the following sections we will describe four identified irrigation triggering routines, including crop model (CM), precipitation delayed (PD), evapotranspiration replacement (ET), and Hydrus 1-D (H).*

11) L135: Why is "(CM)" given here? Same issue with "(H)" in next sentence. The reference/link is not evident from the text.

*This is the abbreviation for the irrigation routine.*

12) Section 2.2.1: I'm a little confused about the differentiation between CM and HM. HM also seems to be linked to Hydrus but not CM? May need a separate description of HM if that is the case or use CM consistently throughout.

*CM and HM are linked. Hydrus uses the outputs from HM.*

13) L150-151: The inputs (e.g., meteorological data, crop biophysical parameters) to the model are not well described here or in Section 2.1.

*We added Table 1 and a data availability section to the manuscript.*

14) L195: "was triggered"

*Corrected*

*L207: triggered*

15) L208: How was daily ETr determined?

*From the HPRCC meteorological dataset.*

16) L222: HM or CM? See previous comment.

*CM and HM are linked. Hydrus uses the outputs from HM.*

17) L243: So are you saying that you used a nondynamic (i.e., the same) LAI time-series for all years? Why not consider inter-annual variations in phenology? Does these descriptions of HM also apply to CM?

*No, just a single LAI time series for all irrigation routines. The LAI time series is on the daily time step and varies from year-to-year. The description will clarified to:*

*L255-256: For each year's growing season we simulated a daily LAI time series using HM.*

18) L244 and L250: The sentence "In addition, HM...." is repeated here.

*Updated and removed the repetition.*

19) L307: There's an issue with the figure numberings. Fig. 5 referred to here is Fig. 6.

*Yes, updated the figure number.*

20) L317: This is not Fig. 6 but Fig. 5.

*Yes, updated the figure number.*

21) L317-323: I'm confused about these numbers, which seem somewhat conflicting. It is stated that both CM and PD are near the historical average. But then it is mentioned that CM is 80 mm lower, the same as ET. In addition, the percentages differ. I also find it difficult to verify these numbers based on the figure. These issues will need to be clarified.

*Agreed, this does need clarification. The slopes are similar but with an offset. The percentages were clarified.*

*L331-332: Both the CM and PD routines reproduce the trend of the historical irrigation amounts but with a low offset (similar slopes).*

22) L323: Fig. 5?

*Yes, will update the figure number.*

23) Section 3.5: Why is ET and PD not mentioned here?

*The don't have a soil consideration within the routine and so soil texture will not have an impact on their numbers. This will be mentioned in the text.*

*L357: Both ET and PD do not have a soil component considered in their routine and as such are not impacted by soil texture.*

24) Section 3.6: In Fig. 7, the CM and ET colors can't be distinguished.

*We will update both colors and line weights for clarity.*

25) L353-354: The historically reported yield should also be plotted on the figure for comparison.

*We only have historical yield for years prior to the study.*

26) L371: Was the 30% reduced irrigation need described/mentioned in the results?

*Updated to up to 115 mm or 30%.*

27) L401-413: This section is a little hard to follow and should be rewritten for better clarity.

*We have made edits for clarity.*

*L426: With regard to soil texture differences in the study area, observed irrigation data indicated no difference between fields in these two texture classes. Similar behavior was seen from the irrigation routine simulations that showed 10% difference for H and 1% difference for CM. We note that given the similar soil texture classes (and thus soil hydraulic parameters) this result is not unexpected. In practice, we are finding that producers are being to adopt precision irrigation techniques (Hedley and Yule, 2009; Hedley et al., 2013). Here, small scale features within a field (e.g. sandy or gravelly areas, underperforming parts of the field, water ways, pivot roads, etc.) can be better managed with the new technology. Therefore, managing fields following 1 dominant soil type (i.e. irrigation-pressure trigger point) may be highly inefficient (Kranz et al., 2014). More refined and consistent soil texture data across arbitrary political boundaries (Chaney et al., 2016) are needed to better account for differences in irrigation water application on the sub-field scale, especially in areas with increasing adoption of precision agriculture technology.*

28) Section 4.4: This section is very brief and would benefit from a much more substantial and elaborate description of the feasibility and limitations associated with the integration of the routines in the LSMs.

*We have expanded this section as requested.*

*L461: The four irrigation routines although biased, capture year-to-year variation in irrigation in Western Nebraska. Given the widespread use of center-pivots we expect the irrigation routines to be appropriate for the HPA and into parts of the eastern USA. Gibson (2016) provides a fuller assessment of irrigation behavior throughout central Nebraska. We note that it is unclear how these routines would behave in areas with center-pivot outside the USA (i.e. Brazil, South Africa, Australia), where energy costs for pumping may be more restricting and drive human-decisions on irrigation. Assessment of these routines in those areas would require further validation.*

*We believe the routines combined with a reasonable bias correction could be easily incorporated into future hyper-resolution LSMs with the above routine descriptions and readily available LSM model output or datasets (see Table 1). Clearly accurate and local precipitation is critical in driving these irrigation routines and capturing producer behavior. This topic deserves more research, particularly and the opportunity to combine low cost in-situ gages with radar and remote sensing products. Additionally, we note the four routines could be run offline in order to provide reasonable guesses of applied irrigation for a given irrigation season. This may be beneficial in representing processes not explicitly considered in LSMs (Kumar et al. 2015), or making future assessments and recommendations about water availability for managers. Finally, the four routines provide reasonable irrigation bounds and more importantly predictions about decreases in irrigation as technology is introduced and adopted in novel areas.*

29) L447: Isn't the 1 km scale often too coarse to resolve field-specific irrigation dynamics?

*Not necessarily for this landscape. The land is partitioned into 0.8 km sections. Often irrigation decisions are made for uniform conditions. Some sub field decisions using precision agriculture are now available but not widely used yet.*

[revised manuscript text omitted]

---

## Author Response (AR2)

Dear Prof. McCabe,

We would like to thank you and the two reviewers for your time and comments regarding our manuscript, titled "A case study of field-scale maize irrigation patterns in Western Nebraska: Implications to water managers and recommendations for hyper-resolution land surface modelling". We have made minor/technical revisions to our manuscript as suggested. You can find our detailed responses to the reviewers' comments (shown in red italics) and the changes we made to the manuscript in the following sections. We have also included a marked up version of the original manuscript.

On the behalf of all coauthors, I am glad that the revised version meets the publication standard of Hydrology and Earth System Sciences (HESS) and inclusion in the Eric F. Wood special issue. Please let us know if there are more questions and comments about the manuscript.

Sincerely,

Justin Gibson

School of Natural Resources

University of Nebraska-Lincoln, USA

**Editor Decision: Publish subject to technical corrections** (31 Jan 2017) by Matthew McCabe
Comments to the Author:
Dear Justin.

I have reviewed the referee reports for your revised manuscript. After implementation of your extensive revisions, I believe that the manuscript provides a much clearer presentation of results and allows the reader to better interpret your findings. As such, I am very pleased to accept your interesting paper, subject to some minor technical corrections. These are largely grammatical in nature, and refer to some additional comments from Referee #2.

Our editorial office will no doubt be in touch with further instructions on how to proceed with publication.

Thank you for your time and effort in developing this thoughtful contribution.

Best wishes,
Matt

*Fantastic! We have made the technical corrections as suggested.*

Report #1

Accept as is.

*Great, thank you.*

Report #2

I'm generally happy with the revisions and only have a few minor technical corrections.

1) You shouldn't capitalize the first letter of all words in the Section titles

*We have made the changes.*

2) Page 16 L342: "management will discussed in.." Should be "will be discussed.."

*We have made the change.*

3) Page 17 L363: Table 1 is reference here but it should be Table 2

*We have made the change.*

4) Page 20 L432: "that producers are being to adopt" - not clear what is meant here.

*We have changed to "producers are beginning to adopt precision…".*

5) Page 22 L476 and Page 23 L493 and L495: "in-situ gages" - should be "in-situ gauges"

*We have made the changes.*

6) Figure 1: I would make the line thickness for the field boundary overlay slightly thinner. Should also describe in the caption what these boundaries actually show.

*We have made the changes. New caption is "Fig. 1: Study area located in western Nebraska with a 1km grid (white lines) overlain on the study site. Black lines show individual field locations where irrigation volumes/depths are obtained from the SPNRD."*

7) Figure 6 is not referenced anywhere in the text!

*Good catch. We have swapped figure 5 and 6 and added the reference to new figure 5 in section 3.2 and changed the other references to new fig. 6.*

8) Figure 3 caption: Weatherstation should be in two words

*We have made the change.*

9) Figure 6 caption: Correct the spelling of "precipipitation"

*We have made the change.*

[revised manuscript text omitted]